# Irrational behavior in *C. elegans* arises from asymmetric modulatory effects within single sensory neurons

Shachar Iwanir[1,2], Rotem Ruach[1,2], Eyal Itskovits [1], Christian O. Pritz[1], Eduard Bokman[1] & Alon Zaslaver[1]

*C. elegans* worms exhibit a natural chemotaxis towards food cues. This provides a potential platform to study the interactions between stimulus valence and innate behavioral preferences. Here we perform a comprehensive set of choice assays to measure worms' relative preference towards various attractants. Surprisingly, we find that when facing a combination of choices, worms' preferences do not always follow value-based hierarchy. In fact, the innate chemotaxis behavior in worms robustly violates key rationality paradigms of transitivity, independence of irrelevant alternatives and regularity. These violations arise due to asymmetric modulatory effects between the presented options. Functional analysis of the entire chemosensory system at a single-neuron resolution, coupled with analyses of mutants, defective in individual neurons, reveals that these asymmetric effects originate in specific sensory neurons.

---

[1] Department of Genetics, Silberman Institute of Life Science, Edmond J. Safra Campus, The Hebrew University of Jerusalem, Jerusalem 9190401, Israel. [2] These authors contributed equally: Shachar Iwanir, Rotem Ruach. Correspondence and requests for materials should be addressed to A.Z. (email: alonzas@mail.huji.ac.il)

*C*aenorhabditis elegans worms exhibit a wide range of innate behaviors, such as chemotaxis toward attractive cues, repulsion from aversive stimuli, and mating[1–3]. These behaviors are typically initiated when amphid chemosensory neurons sense external chemical cues[4]. These sensory neurons, consisted of 12 neuron pairs, can sense and distinguish among a rich set of positive and negative chemical cues, which eventually translate into acute and lasting behavioral outputs, such as attraction and repulsion[4–9].

In total, the neural network of *C. elegans* consists of 302 neurons. This compact network was fully mapped[10], and a myriad of experimental tools, including functional imaging of multiple neurons and neuron-specific genetic manipulations, became available[7,11–14]. Together with the advent of large-scale high-throughput behavioral assays, *C. elegans* worms offer a powerful system to delineate neural-based mechanisms of behavioral outputs. Furthermore, using hermaphroditic *C. elegans* worms bears an important advantage when studying behavior. Although these hermaphrodites maintain variability at the individual level[15–17], the fact that they are isogenic, and grown in the laboratory under the exact same conditions, significantly reduces behavioral variability between individuals.

Of particular interest are innate behaviors (i.e., attraction or repulsion) that are genetically hardwired in the organism. As such, they can be executed in response to a stimulus without prior experience. In particular, *C. elegans* worms chemotax toward food-associated cues, including odorants (e.g., diacetyl and iso-amyl alcohol) and tastants (e.g., NaCl and sodium acetate)[1,2]. The attraction to diacetyl, for example, is mediated primarily by two pairs of chemosensory neurons, AWA and AWC[2,17]. Moreover, the AWA neurons sense diacetyl by the exclusively expressed ODR-10 GPCR[18]. Upon binding, intracellular signaling events activate the AWA neurons, which dictate direct locomotion towards higher concentrations of the chemotactic cue, thus leading the worm to the food source[17,19,20].

The innate capacity to chemoattract towards food cues was presumably shaped over evolutionary time scales to maximize fitness. This raises the possibility that animals will adhere to neuroeconomic-driven theories that postulate rationality in decision-making processes[21–24].

The rational choice theory serves as a classical framework in the study of decision making, with major impacts on various fields, including economy, psychology and evolutionary ethology[25,26]. Within this framework, the fixed utility theory assumes that the rational decision-maker assigns a subjective absolute value (utility) to any action or option. Moreover, these utilities are context-independent such that options can be hierarchically ranked. As a consequence, a rational decision-maker should be consistent when choosing between options. The existence of such a utility function that leads to consistency relies on the axiom of transitivity[26–28].

The transitivity axiom states that if option A is superior to (i.e., preferred over) option B, and option B is superior to option C, then option A must be superior to option C (mathematically, if [A>B & B>C] then A>C)[27]. More-stringent forms impose claims as to the magnitudes of the preferences, so that preference of A-over-C should be greater than either (or both) the A-over-B and B-over-C preferences[29–35].

Two additional principles are central to the fixed utility theory: the principle of Independence of Irrelevant Alternatives (IIA) and regularity. IIA states that the choice between two alternatives is independent of the option-set composition[26,33,36], and regularity asserts that the probability of choosing a given option should not increase when expanding the set of options[33]. However, a myriad of behavioral studies have repeatedly demonstrated that humans, and animals alike, frequently violate these rational decision-making principles[21,23,24,26,29,32,37–47].

In this work, we study rational decision-making paradigms in the worms' innate chemotaxis behavior. We establish *C. elegans* nematodes as a model system for studying rational behavioral decision making, and elucidate the principles underlying worms' violations of rational behavior. Large-scale and hypothesis-free behavioral analyses reveals that *C. elegans* worms violate key rational decision-making paradigms, including transitivity, IIA, and regularity, owing to asymmetric modulatory effects between the presented options. Furthermore, functional analysis of the entire chemosensory system, coupled with behavioral analysis of mutant strains, defective in key sensory neurons, demonstrates that the asymmetric modulatory effects are mediated by specific sensory neurons.

## Results

**Large-scale choice assays reveal violations of transitivity**. In order to investigate worms' rationality, we took advantage of *C. elegans* compatibility with large-scale experiments, and constructed a comprehensive data set of binary preferences between various chemo-attractive stimuli. For this, we placed worms at the center of a petridish, and allowed them to freely choose between two options located at different edges of the dish (Fig. 1a). At the end of the experiment, we calculated the relative chemotaxis index between the pair of options A and B, denoted by RCI(A/B), and which is given by: (no. of worms in A−no. of worms in B)/(no. of worms in A + no. of worms in B). When option B was a null option (buffer-containing option), we denoted it as the Basal Chemotaxis Index, BCI(A) (see Methods).

Overall, the data set consists of 28 options (various doses of seven different stimulants + null option), each tested against all other options, totaling in 378 pairwise comparisons (Fig. 1b, Supplementary Table 1). Notably, analysis of the relative preferences between pairs of stimuli revealed that worms typically (> 85%) preferred the option with the higher BCI, suggesting that worms' innate behavior generally conforms to a hierarchical preference structure.

Next, we asked whether the worms conform to the transitivity rule, a fundamental property of the rational decision-making theory[26,32,35]. For that purpose, we used the data set from the binary choice assays (Fig. 1), and compared the three binary relationships within a triple combination of options. We were able to detect putative triplets that violated all the different types of transitivity (Fig. 2a, Supplementary Table 2). We complemented this discrete-based classification analysis with a parametric approach[28,48,49], assigning a numeric value to the magnitude of intransitivity (Fig. 2a, see Methods). Both analyses revealed that many sets which violated transitivity consisted of the stimulants TT (trimethylthiazole), DA (diacetyl), and IA (isoamyl alcohol) (Supplementary Fig. 1, Supplementary Tables 1–2). Indeed, dose-optimization experiments enabled us to identify a specific dose combination of these three stimuli that robustly led to full intransitivity (Fig. 2b–c, Supplementary Fig. 2). In this set, worms significantly preferred TT over IA and IA over DA, but then preferred DA over TT. Notably, TT was preferred over IA in a direct competition, in spite of its lower BCI (Fig. 2c). Importantly, intransitivity did not arise due to different preference orders which individual animals may have (resulting in a voting paradox, or the Condorcet's paradox[50]), as the small fraction of animals that chose TT in the first choice-assay trial, significantly preferred DA when retested in a second choice assay (Supplementary Fig. 3).

Overall, our large-scale behavioral screen consisted of 3276 triple sets; however, many of the choice comparisons were based on a small number of experimental repeats (Fig. 1). To overcome this, we constructed a noise model, which allowed

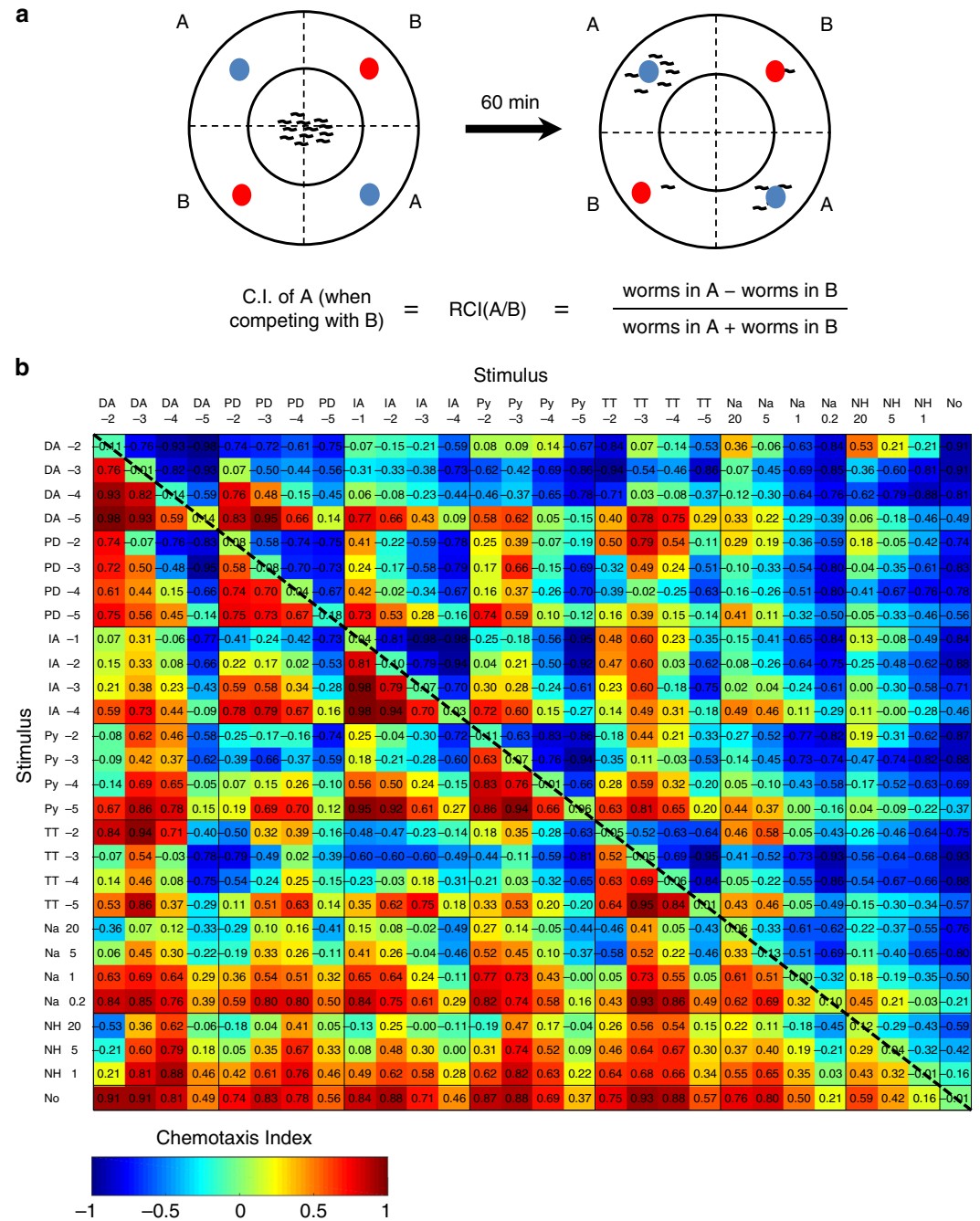

**Fig. 1** A comprehensive pairwise preference matrix. **a** The experimental scheme for the binary choice assay. 100–200 worms, placed in the center of a chemotaxis plate, are free to choose between two chemo-attractive options, A and B, each of which is applied at two opposite quadrants. At the end of the experiment, number of worms in each quadrant is counted and the relative preference towards option A vs option B, RCI(A/B), is computed. **b** All pairwise preference relationships among the 406 binary choice assays (including same option and buffer-only assays). Values inside the entries specify the RCI of the top option relative to the left option. The bottom row depicts preference of each stimulus against a null (buffer) option, denoting the BCIs. Each entry in the matrix is the mean score of 1–5 experimental repeats with typically 2–3 plates per experiment. The full details of the matrix data set, sample sizes, and statistics are provided in Supplementary Table 1. DA, diacetyl; PD, 2,3-pentanedione; IA, iso-amyl alcohol; Py, pyrazine; TT, 2,4,5-trimethylthiazole; Na, NaCl; NH, NH$_4$Ac. 'No' indicates a null (buffer) option. For volatile cues (DA, PD, IA, Py, and TT), numbers in the top and side labels specify the exponent of the 10$^x$ dilutions. For salts, the numbers represent molar concentrations (see methods)

us to provide a more accurate classification of the various triple sets (see methods and Supplementary Fig. 4). We found that 351 triplets could be classified with statistical significance (Fig. 3b), of which 243 triplets were heterogeneous, (i.e., consisted of three different types of options, Fig. 3c). Interestingly, only a small fraction (3%) of these 243 heterogeneous triplets were fully intransitive. Furthermore, violations of

transitivity were more frequent in triplets consisted of odorants only, rather than in triplets consisted of both odorants and tastants (Fig. 3d–e, $p < 0.01$, $X^2$ test). Notably, while our analyses indicate that the fraction of total intransitive cases is ~10–15%, this percentage greatly depends on the specific set of stimuli used to construct the matrix, as well as on the statistical power for inferring (in)transitivity.

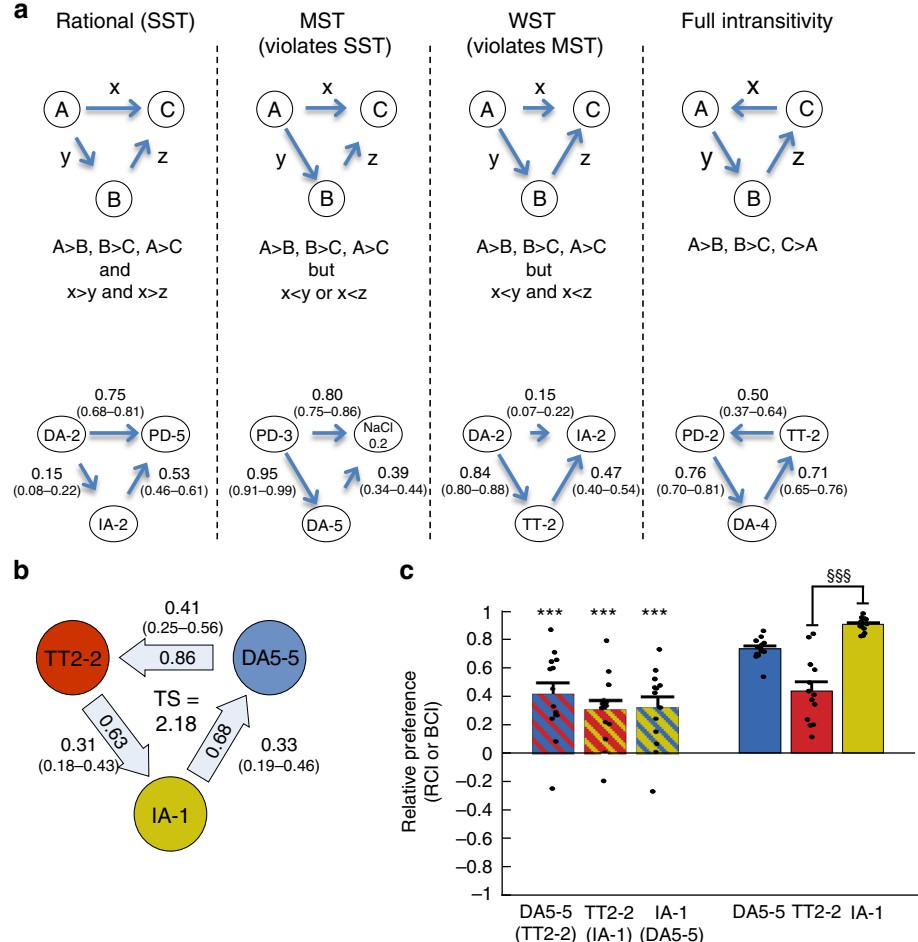

**Fig. 2** *C. elegans* worms violate the transitivity principle. **a** Levels of violations of stochastic transitivity (top, schematic; bottom, examples for putative sets). Arrows indicate the direction of preference, where A → C means that A is preferred over C. For each pairwise comparison, number of experimental repeats is provided in the order of {left pair, right pair, top pair}. Left column: a transitive cycle is maintained (also known as strong stochastic transitivity, SST, see methods, $N = 7,3,2$). Middle columns: in moderate (MST) and weak (WST) stochastic transitivity, the hierarchy order is preserved, but preference magnitude of the direct interaction between the strongest option in the hierarchy and the weakest option is lower than either (MST, $N = 2,6,2$) or both (WST, $N = 3,4,7$) of the two other interactions. Right column: full intransitivity, manifested by a directed cycle within the triple-set options ($N = 3,3,1$). The values next to the arrows indicate the directional preference magnitude (measured as Relative Chemotaxis Index, RCI) with 95% confidence intervals in parentheses (for detailed explanation of calculating confidence intervals see methods). Numbers next to the stimulus specify its dilution (e.g., TT2–2 means TT dilution of $2 \times 10^{-2}$). **b** A robust full intransitivity is found in the triple-set TT2–2/DA5–5/IA-1. Pair scores ($\boldsymbol{log} \frac{P_{(A/B)}}{P_{(B/A)}}$, where $\boldsymbol{P}_{(A/B)}$ is the probability of choosing A over B), are indicated inside the arrows, and the cycle's triple-score (TS) is the directional sum of the Pair Scores (see methods). Values next to the arrows are the RCI's with 95% confidence intervals in parentheses. **c** The three right-side bars denote the basal preference to the stimulus (BCI). The three left-side (striped) bars denote the preference relationships (RCI) of top stimulus vs the stimulus in parenthesis. Notably, for full intransitivity, all three values should be positive (or negative). Note that TT2–2 outperforms IA-1 in spite of its lower BCI. $N = 14$ and $N = 13$ experimental days for the RCI and the BCI measurements, respectively. Scattered dots are the mean chemotaxis indexes of plates in each experimental day. §§§ $p < 0.001$ (t test). *** indicates means are larger than 0 at $p < 0.001$, ANOVA. Error bars denote s.e.m.

Together, coupling our comprehensive choice-assay data with a rigorous analytical approach facilitated a robust platform to identify intransitivity of an innate behavior in *C. elegans* nematodes. These analyses indicated that although *C. elegans* worms generally conform to the transitivity rule, they genuinely violate transitive relationships when required to choose between specific combinations of stimuli.

**Intransitivity arises from asymmetric cross-modulatory effects**. We next dissected the functional interplay between DA, TT, and IA, the three stimuli leading to robust intransitivity (Fig. 2b). Specifically, we speculated that modulation of chemo-sensation to a given stimulant by the presence of another stimulant may underlie the observed violations of transitivity. For this, we

systematically studied how a homogeneous background of one of these stimulants affected the attraction toward another (Fig. 4a). We found that the presence of a DA background significantly affected worms' preference toward TT in a concentration-dependent manner (Fig. 4b top, Supplementary Fig. 5a top): low DA background levels strongly attenuated attraction towards TT, and more strikingly, moderate levels of background DA completely reversed the response to TT from clear attraction to robust repulsion. Interestingly, in the reciprocal experiment, a TT background did not affect worms' attraction towards DA (Fig. 4b bottom, Supplementary Fig. 5a bottom), suggesting the existence of asymmetric cross-modulatory effects between the two stimuli. Similarly, we found a mild asymmetric cross-modulatory effect between IA and TT (Fig. 4c, Supplementary Fig. 5b). These

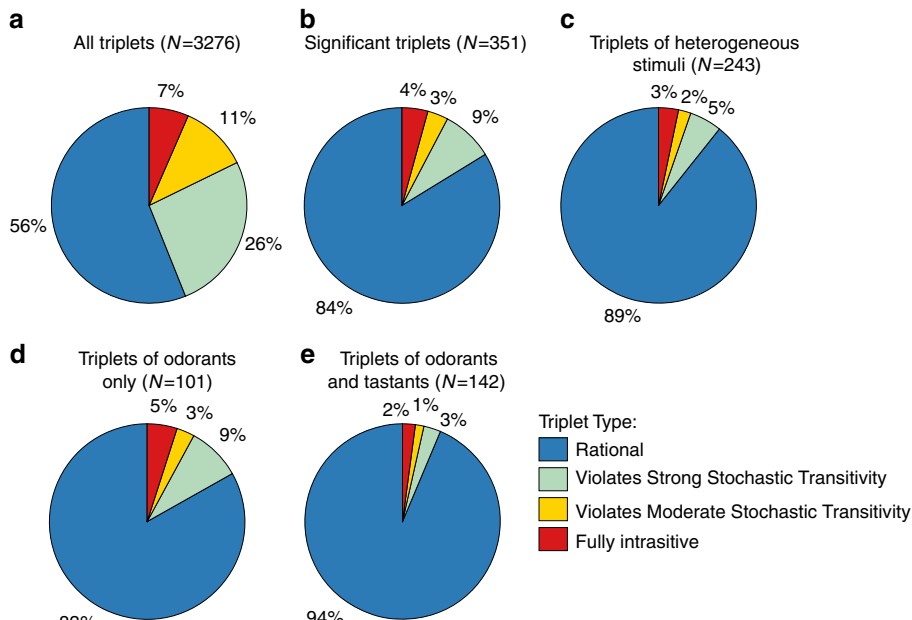

**Fig. 3** Intransitivity is more prevalent when considering options sensed by a single modality. **a** Classification into (in)transitive types of all possible 3276 triplets. **b** Classification of triplets for which significance could be assigned (confidence interval is > 95%, see methods for calculation details). **c** Out of the 351 significant triplets, shown are the classifications of heterogeneous triplets (e.g., consisted of three different stimuli). **d**–**e** Of the 243 heterogeneous triplets in **c**, 101 were consisted of odorants only **d**, and 142 were consisted of at least one tastant and one odorant **e**. The proportion of triplets that violate transitivity (of any type) in **d**, is significantly larger than in **e**, indicating that intransitivity is more common when considering odorants only than a mixture of odorants and tastants ($P = 0.009$, $X^2$ test, $N = 243$)

observations establish that the presence of one stimulus can modulate the innate attraction (preference) towards another stimulus without having a similar reciprocal modulation. Of note, asymmetric reciprocal modulations between options are essential for violating transitivity, as symmetrical modulations will cancel each other out, resulting in overall same preference relationships.

**Calcium imaging reveals cross-modulatory effects in single neurons**. Next, we capitalized on the compatibility of *C. elegans* worms with quantitative measurements of neural activity, and asked whether the asymmetric behavioral phenotypes may be explained by asymmetric coding dynamics at the sensory layer of the animal. In that respect, the chemosensory system of *C. elegans* nematodes has been extensively studied and individual neurons were shown to respond to a range of chemical cues, including DA, TT, and IA[19,51,52]. For this, we generated an integrated transgenic strain in which the genetically encoded calcium indicator GCaMP is expressed in all chemosensory neurons, thus allowing to measure simultaneous activity from virtually all chemosensory neurons at a single-neuron resolution (Fig. 5a–b).

To mimic the gradual changes in the concentrations that worms experience during the chemotaxis choice assays (Figs. 1, 4), we used a custom-made microfluidic setup[17] that allows exposing animals to smooth increasing and decreasing concentrations of the stimuli (Fig. 5c–d, top panels). Specifically, we focused on the pair of cues DA and TT, which exhibited the most prominent asymmetric cross-modulatory effects (Fig. 4b, Supplementary Fig. 5), and measured neural dynamics under four different conditions: Sinusoidal gradients of TT with and without a background of DA (Fig. 5c), and the reciprocal condition of sinusoidal gradients of DA with and without a background of TT (Fig. 5d).

We identified several chemosensory neuron types whose mean activity significantly changed after the gradient onset

(Supplementary Fig. 6). Of these, activity of ASK, ADL, and ASH was owing to their response to the blue light[52], and indeed their activity starts high from the beginning of the imaging (and not from the gradient onset) and continuously decays over time. ASI also exhibited a mild reduced fluorescence that was not timed with the gradient onset, and is probably attributed to bleaching.

Four neuron types (AWA, AWB, AWC, and ASJ), however, showed a significant differential activity synced to the gradient onset in at least one of the conditions (Fig. 5). The AWA pair of neurons strongly responded to either TT or DA, and this response was significantly diminished when the other stimulant was present at the background (Fig. 5e–f). AWB and ASJ neurons responded to TT only, with no clear effect of a DA background (Fig. 5i–l). The AWC neurons showed variable responses (Fig. 5g–h): their activity mildly decreased in response to rising concentrations of either TT or DA, but significantly increased only when DA was present in the background and when TT was applied, but not in the reciprocal condition (Fig. 5g). Interestingly, AWA and AWC are known to control chemotaxis behavior: AWA activity promotes forward locomotion while AWC activity stimulates turning events that often reorient the animal to a new trajectory[17,19].

Together, these results indicate that sensory neurons differentially respond to the reciprocal conditions of DA and TT, suggesting that the differential modulation observed at the behavioral level may be originating at the sensory layer of the animal.

**Cross-modulatory effects in AWA may lead to irrational behavior**. Functional analyses of neural dynamics revealed four chemosensory neuron types (AWA, AWB, AWC, and ASJ) that participate in coding the two reciprocal DA and TT conditions. Of these, responses of AWA and AWC to one cue were cross-modulated depending on the presence or the absence of the

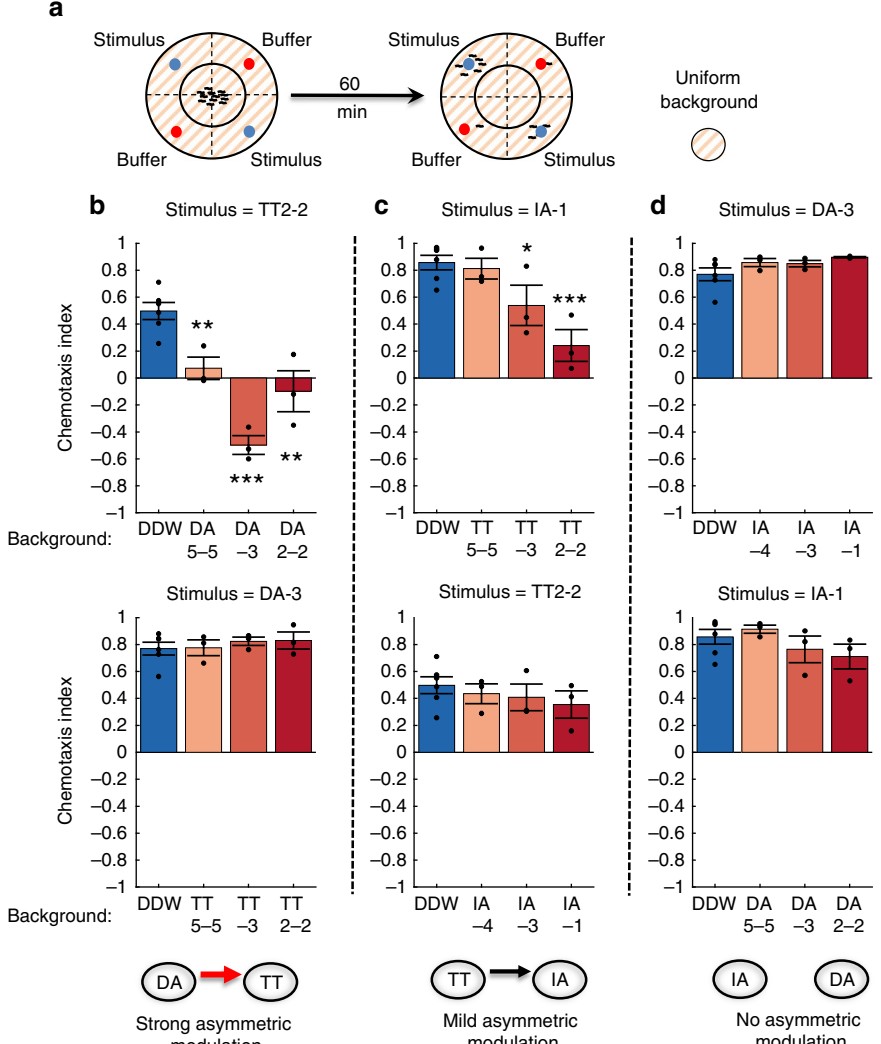

**Fig. 4** Asymmetric cross-modulatory effects between pairs of stimuli. **a** Scheme of the experimental arena. Prior to the experiment, the background stimulant was uniformly applied to a filter paper placed on the internal face of the plate cover. **b**–**d** Effects of various background concentrations (shown horizontally) on attraction toward a target stimulus (noted at the top of each panel). Note the asymmetric cross-modulatory effects between DA & TT, and between TT & IA. Numbers next to the stimulus specify its dilution. $*p < 0.05$, $**p < 0.01$ $***p < 0.001$ (two-sample $t$ test). For DDW background, $N = 6$ experimental days; for other backgrounds, $N = 3$. Each experimental day consisted of 2–4 plates of each condition. Scattered dots are the mean chemotaxis indexes of plates in each experimental day. Error bars denote s.e.m.

second cue (Fig. 5e–h). AWA and AWC had been shown to mediate attraction towards DA[2,17,19]. Likewise, AWA neurons respond to the stimulant TT[13], and behavioral assays showed that both AWC and AWA are key for chemotaxis towards TT[2,53]. To elucidate the functional roles that AWA and AWC neurons play in mediating chemotaxis under these conditions, we used mutant strains in which these neurons are defective: (I) guanylyl cyclase *odr-1* mutant, in which AWC-mediated sensation is postulated to be completely defective[2,53–55]; (II) an AWA-specific nuclear receptor *odr-7* mutant, a gene crucial for the proper AWA neural fate determination[53,56–58].

First, we analyzed the chemotaxis of these strains towards TT in the presence or the absence of a DA background (Fig. 6a). In line with previous findings (Fig. 4), wild-type worms were repelled from TT in the presence of a DA background. A similar behavior was observed in animals' defective in AWC neurons, indicating that AWC neurons are dispensable for mediating both attraction and repulsion under these conditions. In contra-distinction, AWA neurons were essential for mediating attraction toward TT, as animals with defective AWA neurons were repelled

from TT even in the absence of background DA, suggesting that worms have a basal, AWA-independent, repulsion from TT that is masked by functional AWA activity. When assaying chemo-taxis to DA in the presence or the absence of a TT background, we found that AWA contributes to DA attraction, as worms lacking functional AWA neurons are significantly less attracted to DA (Fig. 6b).

To corroborate these results, we repeated the chemotaxis assays with an additional mutant strain, in which the AWC neurons were genetically ablated, and obtained similar results (Fig. 6c). Interestingly, a double mutant, defective in both AWC and AWA neurons, showed a mildly lower retraction than the single AWA-defective mutant (*odr-7*), suggesting that AWC neurons mediate a low basal retraction from TT. These findings, together with the results obtained by neuroimaging (Fig. 5), suggest that a background DA, sensed via the AWA neurons, modifies AWA activity and leads to repulsion from TT.

To further delineate the mechanism by which DA affects AWA-mediated TT sensing, we assayed chemotaxis in two independent null mutant strains of the *odr-10* gene, encoding an

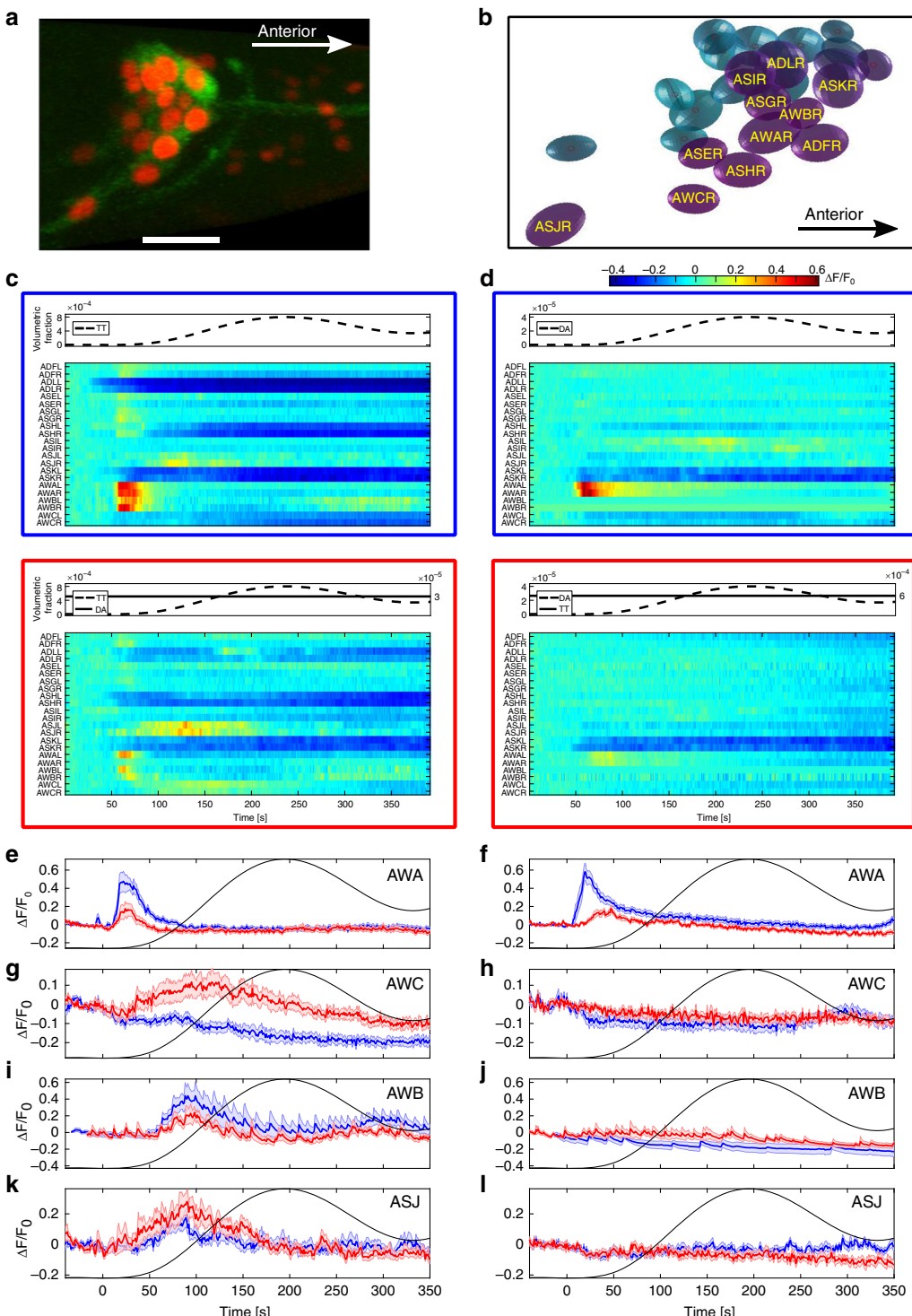

AWA-specific DA receptor[12,13,18]. Both *odr-10* mutants displayed normal attraction to TT, but in contrast to wild-type animals, this attraction remained intact and did not switch to repulsion in the background of DA (Fig. 6d). These findings indicate that DA modulates the attraction to TT through its action on the ODR-10 GPCR in the AWA neurons. Moreover, it suggests that TT sensation in the AWA neurons is mediated through a different receptor than ODR-10, as *odr-10* mutants were still attracted to TT (Fig. 6d).

The neuroimaging analyses (Fig. 5), together with the results obtained by behavioral assays of mutant strains (Fig. 6a–d), offer

a model whereby the DA-mediated TT repulsion may lead to asymmetric modulation and irrationality (Fig. 6e). In the absence of DA, TT is sensed by a, yet to be identified, receptor on the AWA neurons, whose strong activation mediates attraction (Fig. 5e, Fig. 6a, c). Additional sensory neurons, including AWC, mediate a weak repulsion from TT, such that the overall effect is attraction (Fig. 6a, c, e left panel). When DA is present, it is sensed by the AWA neurons through the ODR-10 GPCR (Fig. 6d), leading to a decreased activity of the AWA neurons in response to TT (Fig. 5e). This translates to the observed AWA-mediated repulsion (Fig. 6c). In parallel, AWC and other sensory

**Fig. 5** Calcium imaging of chemosensory neurons reveals modulated neural responses. **a** An overlay image of the GCaMP (green) and the nuclear NLS-mCherry (red) signals of the transgenic strain used for imaging all chemosensory neurons. Bottom scale bar marks 10 μm. **b** A snapshot from the analysis pipeline showing the identified neural nuclei (image is not to scale). Fluorescent intensities were extracted from the nuclei spheres. **c–d** Mean fold-change activity of individual chemosensory neurons. **c** Worms exposed to a sinusoidal gradient of TT in the absence (top, $n = 10$ animals) or the presence (bottom, $n = 7$ animals) of a DA background. **d** Worms exposed to a sinusoidal gradient of DA in the absence (top, $n = 10$ animals) or the presence (bottom, $n = 8$ animals) of a TT background. **e–l** Mean fold-change activity of the four neuron types that significantly responded to a chemical gradient in any of the four conditions. **e, g, i, k** Responses to TT gradients in the absence (blue) or the presence (red) of a DA background. **f, h, j, l** Responses to DA gradients in the absence (blue) or the presence (red) of TT background. As right and left lateral neurons showed similar responses, presented are the means of both neurons. As activity of AWB neurons was occasionally masked by the strong AWA activity, responses shown in **l** and **j** were recorded in a separate line expressing GCaMP exclusively in AWB (see methods). Shaded areas denote s.e.m. Mean AWA activity in the first 150 s of the gradient was significantly weaker for both DA and TT in their reciprocal background presence compared with their respective null background conditions (blue vs red curves in **e**, **f**, ranksum test, $p = 0.016$ and $p = 0.006$, respectively). Mean AWC activity in response to a TT gradient was significantly elevated during gradient application in the presence of a DA background but not in its absence (blue vs red curve in **g**, ranksum test, $p = 0.0015$). In **e–l**, the number of neurons imaged with and without background (red and blue, respectively) is: (**e**: 10,20); (**f**: 17,18); (**g**: 12,19); (**h**: 11,15); (**i**: 7,5); (**j**: 5,8); (**k**: 14,20); (**l**: 13,19)

neurons continue to mediate the basal repulsion from TT, independent of the DA background (Fig. 6c). Thus, the combined activity of these neurons shifts the worm preference from strong attraction to repulsion (Fig. 6e right panel).

**Trans-modulatory effects lead to violation of IIA and regularity.** We next hypothesized that the observed asymmetric effects between specific stimuli will also lead to violations of other key rationality paradigms, namely, the Independence of Irrelevant Alternatives (IIA), and regularity: IIA posits that the preference ratio between two alternatives should be independent of other available options[26,33,36], whereas regularity asserts that the probability of choosing an option should not increase when adding a new option to the existing set of options[33].

To study whether *C. elegans* innate behavior also violates these rationality paradigms, we adapted the chemotaxis assays to study preferences among three options when presented simultaneously (Fig. 7a). We found that IA was preferred over DA (by approximately twofold) when buffer was used as the third option (Fig. 7b). Interestingly, replacing the buffer with TT not only changed this ratio, but it actually switched between the preferences, such that DA became preferred over IA (by approximately twofold), thus violating IIA. Furthermore, expanding the set of options from two (IA/DA) to three (IA/DA/TT) increased the overall fraction of worms preferring DA (Fig. 7c), thus violating the regularity paradigm. Importantly, the relative preference to TT was poor (~10%, Fig. 7c) conforming to a strict definition of "irrelevance" of the added option.

## Discussion

In this study, we used *C. elegans* nematodes for studying rational decision making, and delineated the principles that lead to its irrational behavior. Notably, we demonstrated violations of several rational behavior paradigms (namely, transitivity, IIA, and regularity), all within the scope of a single experimental framework.

First, we constructed a comprehensive pairwise preferences matrix (Fig. 1), which allowed us to uncover sets of options that consistently violated the transitivity rule (Fig. 2). Focusing on a specific triple-set of attractants (DA/TT/IA), we found that asymmetric cross-modulatory effects underlie the observed intransitivity (Fig. 4). Functional analysis of the entire chemosensory system at cellular resolution revealed the individual neurons that encode the different stimuli (Fig. 5), and behavioral analyses of mutants, defective in these sensory neurons, allowed us to propose a simple parsimonious mechanism that may explain the observed intransitivity (Fig. 6e). In the absence of a DA background, animals are attracted toward TT, and AWA

neurons dominate over other neurons to mediate this attraction. In the presence of a background DA, AWA activity switches to mediate repulsion rather than attraction. Concomitantly, AWC, and possibly additional neurons, also contribute to the repulsive response.

Importantly, such a behavioral switch is not observed in the reciprocal condition where animals chemotax towards DA in the background of TT. This is exactly what defines the asymmetric cross-modulatory effect between options and which ultimately leads to intransitivity. In fact, to observe intransitivity within a triple-set of options, it suffices that a single pair of options will exert asymmetric cross-modulatory effects (Fig. 8a). Notably, in our triple-set of options, we identified asymmetries between DA and TT and between TT and IA (Fig. 4). Together, they additively contributed to the formation of a robust full intransitive triple-set (DA → TT → IA → DA, Fig. 2b).

To construct the comprehensive choice matrix (Fig. 1), we used stimuli that are known to attract *C. elegans* worms[4]. Moreover, some of these stimuli had been shown to be secreted by bacteria, upon which the worms feed[59,60]. Although the concentrations used in our assays may not reflect the typical concentrations found in the natural ecological niches, they still served as potent attractors; thus, the irrationalities reported herein are found in the context of the worms' evolutionary-shaped innate chemotaxis behavior.

It was previously shown that the presence of a background stimulus can interfere with chemotaxis towards another stimulus[2]. Here, we found that the presence of an attractive background stimulus not only affected attraction, but actually reversed the preference from attraction to repulsion. *C. elegans* worms may switch preferences from attraction to repulsion when a stimulus concentration increases beyond a certain threshold[2,51,61]. These repulsive responses often trigger the nociceptive neurons AWB and ASH[51]; however, our calcium imaging data indicate that activity of these neurons is not modulated between the reciprocal conditions. Instead, we found that AWA, and possibly AWC, are the major players contributing to the DA-mediated TT repulsion. Moreover, the data suggest that TT and DA are sensed by two different receptors on the AWA neurons (Fig. 6). Thus, the behavioral switch, and concomitantly the irrational behavior, may largely result from input integration occurring within a single type of neurons, AWA, where each input (stimulus) is sensed by its own cognate receptor. Interestingly, such a dual role for a single-neuron type was suggested for the AWC neurons, which may mediate both attraction and repulsion from butanone via differential synaptic signaling[62].

Stimuli converging onto the same sensory neurons may also underlie the other instances of intransitive triplets that we observed through the comprehensive pairwise choice assays

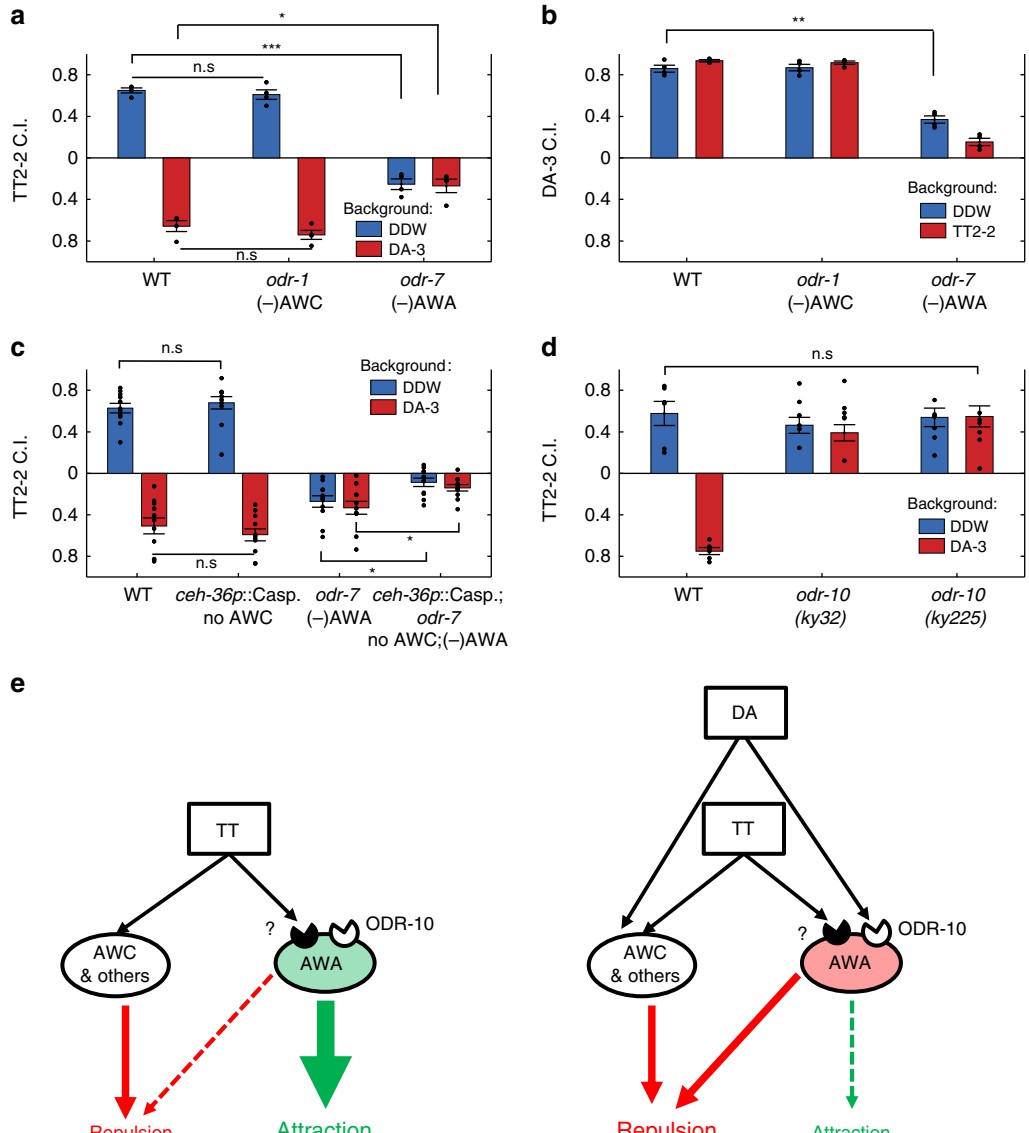

**Fig. 6** Sensory modulations, primarily within the AWA neurons, underlie the irrational behavior. **a**, **b** Chemotaxis index towards TT ($2 \times 10^{-2}$, **a**) and DA ($10^{-3}$, **b**) in the presence or the absence of a background of DA ($10^{-3}$, **a**), or a background of TT ($2 \times 10^{-2}$, **b**). For all groups, $N = 4$ experimental repeats (days). **c** Chemotaxis index towards TT ($2 \times 10^{-2}$) in the presence or the absence of a DA background ($10^{-3}$). $N = 11$ experimental repeats for all groups. **d** Chemotaxis index towards TT ($2 \times 10^{-2}$) in the presence or absence of a DA background ($10^{-3}$). $N = 6$ experimental repeats for all groups. Experiments in each of the panels **a**–**d** were conducted on the same days, and each experimental day is the average of 2–4 plates assayed for each condition and/or strain. Scattered dots are the mean CI of plates in each experimental day. ($-$) in front of a neuron name indicates that the neuron exists but is disfunctional. *$p < 0.05$, **$p < 0.01$, ***$p < 0.001$ (paired $t$ test). Error bars denote s.e.m. **e** A proposed model which may explain intransitivity based on our data. Left, when TT is the sole attractant, and in the absence of DA in the background, the prevalent attraction is mediated by AWA, whereas AWC and other neurons mediate negligible repulsion. Right, in the presence of a DA background, DA is sensed by the ODR-10 receptor on the AWA neurons. The action of DA presumably diminishes the AWA-mediated attraction towards TT (at these concentrations) and augments the repulsive response. Dashed arrows indicate the possible modulated effects that AWA neurons exert

(Fig. 1–3). In support of this, we found that intransitivity is significantly more common when considering same-modality stimuli (e.g., odorants only, Fig. 3c–d), which are sensed by the same sensory neurons[2]. Together, modulatory effects of stimuli, that change neural activity and behavioral outputs, may eventually lead to irrational decision making in *C. elegans*.

Although focusing on transitivity, we speculated that additional paradigms of irrational behavior may be violated owing to differential modulatory effects between options. Following the finding of modulatory effects that TT exerts on IA (but not on DA, Fig. 4b–c), we were able to design experimental conditions that will exploit this asymmetry to demonstrate violations of IIA

and regularity (Fig. 7). In this case, trans-modulatory effects, rather than cross-modulatory effects, lead to violations of these paradigms, where option C (TT) modulates the preference to option A (DA) differently than it modulates the preference to option B (IA) (Fig. 8b).

Interestingly, analogous trans-modulatory effects that lead to irrational behavior were observed in humans as well as in other vertebrate and invertebrate animals[39,46,63,64]. These are known as the asymmetric dominance effect, or the decoy effect, a form of context-dependent choice in which the probability of choosing one of two options is impacted by the introduction of a third, weaker option[65]. According to this effect, adding to a set of

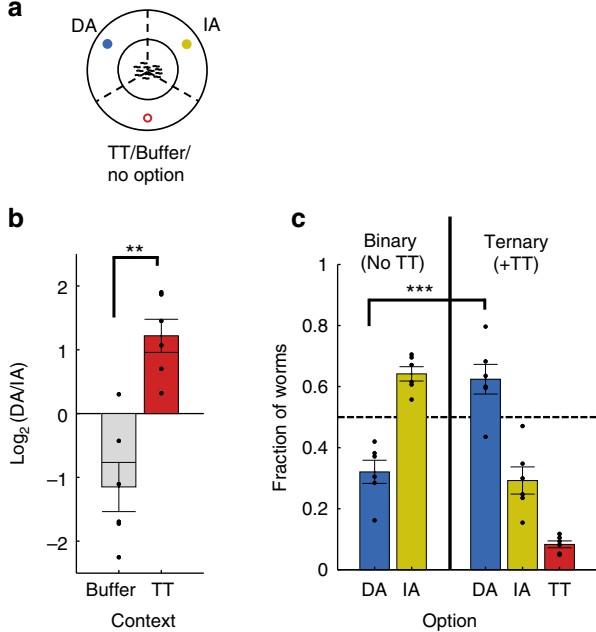

**Fig. 7** *C. elegans* worms violate the rationality paradigms of IIA and regularity. **a** A diagram of the ternary choice assay. **b** Violation of Independence of Irrelevant Alternatives (IIA). For studying IIA, we compared preferences between two competing options (DA $7 \cdot 10^{-5}$ and IA $10^{-1}$) where the third option was either TT $10^{-2}$ or buffer. The identity of the third option (TT vs buffer) changed the preference ratio between DA and IA, thus violating the constant-ratio rule[28]. In fact, it actually switched the identity of the preferred option, thus violating also choice independence of alternatives. *Y* axis is the log ratio between worms that chose DA and worms that chose IA. \*\**P* < 0.005, significance of difference between buffer and TT as a third option (paired *t* test). Importantly, both log ratios are significantly different than 0 (*p* < 0.05, *t* test), indicating that presenting TT, rather than buffer as a third option, switches DA/IA preference from IA to DA. **c** Violation of regularity. Shown are the fraction of worms selecting each option in binary and ternary choice assays. Note that expanding the set by adding TT not only increased preference towards DA but caused this preference to actually exceed 50%. \*\*\*significance of change by set expansion with *P* < 0.001 (paired *t* test). *N* = 6 experimental days for all conditions. In all panels, each experimental day is the average of 2–4 assayed plates. Scattered dots are the mean measurements in each experimental day. Error bars denote s.e.m.

options, A and B, a third option C that is asymmetrically dominated by option A (but not by option B), will increase the preference of choosing A over B. In humans, this effect had been extensively documented in various fields, including marketing and consumption[66], policy decisions[67], and partner preferences[68].

Obviously, mammalian brains are organized completely differently than the *C. elegans* neural network, and irrationality may arise owing to complex integrated dynamics within different brain regions, rather than in individual perceptual neurons. Yet, it is possible that the same principles of asymmetric cross- or trans-modulatory effects between options may serve as the common underlying basis that leads to irrational behavior.

In summary, here we revealed that asymmetric modulatory effects between options underlie irrational decision making in *C. elegans* worms. In particular, we were able to demonstrate that similar principles lead to violations of transitivity, IIA, and regularity, all within the scope of a single experimental framework. As no single existing theory can account for all types of rationality violations[26], our findings may support new understandings of the processes that lead to irrational behaviors.

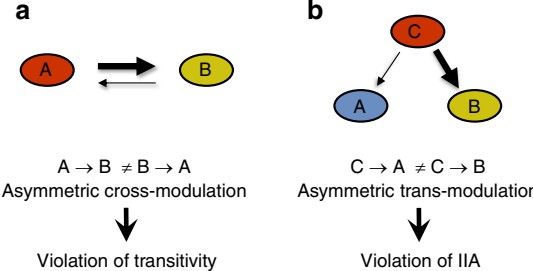

**Fig. 8** Asymmetric modulatory effects between options underlie irrational decision making. **a** Asymmetric cross-modulatory effects between two options (out of a set of three options), where option A affects preference towards option B differently than option B affects preference towards option A ($A \rightarrow B \neq B \rightarrow A$) can explain violation of the transitivity principle. **b** Asymmetric trans-modulatory effects, where option C differently affects preferences towards options A and B ($C \rightarrow A \neq C \rightarrow B$), may explain violation of Independence of Irrelevant Alternatives (IIA) and regularity

## Methods

**Strains used in the study**. The following strains were used: wild-type strain N2, CX32 *odr-10(ky32)*, CX3410 *odr-10(ky225)*, CX2065 *odr-1(n1936)*, CX4 *odr-7(ky4)*, PY7502 *oyIs85[ceh-36::TU#813; ceh-36::TU#814; srtx-1::GFP; unc-122::DsRed]*, ZAS327 *odr-7(ky4); oyIs85[ceh-36::TU#813; ceh-36p::TU#814; srtx-1::GFP; unc-122::DsRed]*, PS6384 *pha-1(e2123ts)*; *syEx1245[str-1::GCaMP3 + PHA-1]*, ZAS280 *azrIs347[osm-6::GCaMP3, osm-6::NLS-mCherry + PHA-1]* injected into *pha-1 (e2123ts)*, X-ray integrated and backcrossed 10 times with N2. All strains were maintained and grown according to standard protocols[69,70].

**Binary choice assays**. Assays were performed on bleached, synchronized young adult N2 worms, 65–70 h post bleaching. Worms were grown on standard NGM plates, washed thoroughly with CTM (5 mM potassium phosphate, pH 6.0, 1 mM CaCl₂, 1 mM MgSO₄) to remove bacteria, and kept food-deprived in CTM for 45–60 min before loaded on the assay plates. Assays were performed in 9 cm covered CTX plates (25 mM KH₂PO₄, pH 6.0, 1 mM CaCl₂, 1 mM MgSO₄, 1.7% agar).

To minimize the effect of possible drifts resulting from loading or from environmental gradients, we applied a quadrant assay-plate design[71], in which the plate was divided into four 90° radial slices, with each pair of opposing quadrants containing the same option (Fig. 1a). In addition, repeats of the same group were differentially rotated to heterogeneously cover an entire circle (e.g., if three repeats were used, plates were 0, 120, and 240° rotated). Within-experiment and between-plate variability was low even when the distribution of worms between the two same-plate, same-treatment opposing quadrants was variable and uneven, suggesting that a bias toward one side was reasonably balanced by the negative bias to the other. As the experiments involved many treatment combinations, often at high concentrations, we applied three means of verification that there is no cross-effect between plates: first, we tested cross-plate chemotaxis in uncovered plates and found it minimal. Second, we combined all tested combinations in plate towers (stack of plates) by repeats, placed the towers at constant (~ 50 cm) spaces, and found no detectible biases or differences in the measured indices between repeats. Third, we occasionally included a no-treatment plate inside each tower, and found no bias in worm distribution in these plates, ruling out significant in-tower cross plate effects.

Stimuli were applied at a distance of 3.5 cm from the plate center, and a 4.95 cm equidistance from the two neighboring rival treatment application points. Volatile substances were applied just before worm loading in a 15 µl drop of the specified dilution in CTM supplemented with 117 mM NaN₃ to immobilize the worms upon reaching the stimulus origin. Soluble stimuli followed a double application protocol as described in ref. [72], with two 5 µl drops of the specified concentration applied 20–24 h and 4–6 h prior to worm loading, and 15 µl of 117 mM NaN₃ in CTM added just before worm loading. Worms were loaded to the center of the plate in a 7 µl drop. Typically, 100–200 worms were loaded to each plate. At these numbers, the effect of the exact worm number on the accuracy of the measured index is negligible.

Experiments were performed at room temperature (~ 20 °C). Assays were allowed to run for at least 60 min., a sufficient time to immobilize essentially all worms on the plate. The number of worms in each quadrant was counted by eye under a stereoscope. Worms found within a radius of 1.9 cm from the center (inner circle in Fig. 1a) were usually < 1% of total number of worms and were not included in the total worm count, so the combined fractions of the two options always summed up to 1. Typically, > 95% of the worms reached to within 1 cm from one of the options, indicating sufficient robustness of the assay results and independence of factors such as NaN₃ concentration.

**Construction of the binary preference matrix**. The preference matrix set was composed of 28 options (stimuli). Volatile cues were tested at four serial 1:10 dilutions ($1:10^2$–$1:10^5$). NaCl was applied at four doses: 4 drops of 5 M, 1 drop of 5 M, 1 drop of 1 M, and 1 drop of 0.2 M. For $NH_4Ac$, we used three doses: four drops of 5 M, 1 drop of 5 M, and 1 drop of 1 M. The condition marked as 20 in Fig. 1b is composed of four drops of 5 M. Volume of each drop was 5 μl. Data were collected during 58 experiment days, totaling in 5670 experimental plates. Typically, a sub-matrix covering all dose combinations of a pair of substances (i.e., a "box" in Fig. 1b) was tested on the same experimental day, with two–four repeats of each pair of combinations. For the exact number of recording days and measured chemotaxis indexes of each binary interaction, see Supplementary Table 1. Overall, the matrix consists of 3276 possible triple sets (choose 3 out of the 28 stimuli without replacements, $28!/(25! \times 3!)$).

Of note, The BCIs of all tested stimuli were positive, confirming their attractiveness, and increased dose-dependently across 2–3 orders of magnitude for volatiles and 1–2 orders of magnitude for salts. Order of potency for the volatile stimulants was TT > DA≈PD≈Py > IA. Preference relationships in choice sets comprised of two dilutions of the same stimulant were always in favor of the higher dose. When different stimulants were used in the choice assay, worms typically preferred the option with the higher BCI.

**Choice assays with a constant background**. For background assays (Fig. 4), we applied the background stimulus on 85 mm No. 1 Whatman filter paper discs placed on the internal side of the Petridish lid. Background was applied at 1 ml volume, which allowed uniform distribution of the liquid throughout the disc. To approximate worm cumulative experience of the background stimulant to its experience as an option, we used a 33 × dilution of the corresponding option in the assay drops. This balances the total plate volume ratio between the drops (15 μl × 2 drops) and the background (1 ml). Note that background dilutions indicated in the figures refer to the parallel dilution as an option (e.g., an indicated 2–2 background dilution refers to 1 ml of $6 \times 10^{-4}$ dilution). Background dilutions were performed in DDW, and control groups of these experiments included a filter paper similarly absorbed in DDW with no stimulant (which essentially displayed similar CIs to that of the no-background assays (compare Fig. 2c with Fig. 4). For DA and TT backgrounds, we used the equally spaced dilutions of $2 \times 10^{-2}$ (2–2; high), $10^{-3}$ (−3; moderate) and $5 \times 10^{-5}$ (5–5; low), covering an overall × 400 concentration range. For IA, we used dilutions of $10^{-1}$, $10^{-3}$, and $−10^{-4}$, thus covering a × 1000 range of concentrations.

**Three (ternary) option assays for studying IIA and regularity**. Stimuli application and worm loading were performed similar to the binary choice assays, except that the plates were divided to three 120° tertiants, and each option was applied to only one point. To overcome possible spatial biases, we rotated the plate repeats as described for the binary choice assay. For the IIA assay (Fig. 7b), the third option contained buffer + $NaN_3$ which, as expected, was irrelevant as a choice compared with the other options (had a mean CI of 0.04 across all experiments), similarly to the binary basal chemotaxis assays. For studying regularity (Fig. 7c), only two tertiants were loaded as options (denoted as no option in Fig. 7a).

**Analysis of the behavioral choice assays**. The formula to calculate the Chemotaxis Index (CI) of option A is[2,71]:

C.I.(A) = (no. of worms in Q1 + Q3−no. worms in Q2 + Q4)/(no. of worms in Q1 + Q2 + Q3 + Q4)

Where Q1 and Q3 are the quadrants containing option A, and Q2 and Q4 are the quadrants containing the alternative option B (Fig. 1a).

In order to define utilities and preferences of a given option A in various binary contexts, we defined the following terms:

Basal chemotaxis index of A [BCI(A)]−preference of option A vs the null option (control, DDW, or buffer). This value also served as a measure for basic utility. BCI(A) = C.I.(A) where the null option occupies the other two quadrants.

Relative chemotaxis index of A vs B [RCI(A/B)]−preference of option A vs an alternative option B. RCI(A/B) = C.I.(A), where option B occupies the other two quadrants. Note that RCI(A/B) = - RCI(B/A).

**Estimation of the standard deviation**. When constructing the binary preference matrix, most binary choice preferences were estimated based on 1–4 experimental between-day repeats. In order to better estimate the variance in worm's preferences (RCI, BCI), we chose 46 different pairs of stimuli (consisted of odorants, tastants, the buffer null option, and which span various magnitudes of CI's), and assayed them on at least five different days with ~ 4 plate replicas on each of the different days. Analysis of these experimental repeats revealed that variance between days was significantly higher than the variance within the same day experiments (Supplementary Fig. 4a). We therefore used the between-days variance to assess the CI variance. In addition, we found a strong negative correlation between the CI absolute value and the between-days standard deviation ($R = −0.63$, $p < 10^{-5}$, Supplementary Fig. 4b). Consequently, we used this linear regression to estimate the day-to-day standard deviation for pairs of stimuli for which we had less than five experimental repeats on different days.

**Triple set analysis (transitivity)**. Consistency of choices within binary combinations of three options was analyzed using two complementary approaches. The first, the ordinal approach, follows the traditional classification of transitivity levels[49]:

We define $P_{(A/B)}$ as the probability of choosing A given the two choices A and B. If option A is preferred over option B, such that $P_{(A/B)} \geq 0.5$, and option B is preferred over option C ($P_{(B/C)} \geq 0.5$), then:

1. Strong stochastic transitivity (SST): $P_{(A/C)} \geq \max[P_{(A/B)}, P_{(B/C)}]$
2. Moderate stochastic transitivity (MST): $P_{(A/C)} \geq \min[P_{(A/B)}, P_{(B/C)}]$
3. Weak stochastic transitivity (WST): $P_{(A/C)} \geq 0.5$
4. Full intransitivity: $P_{(A/C)} < 0.5$

We used the estimated/measured standard deviation to classify the triplet type using a bootstrap method: for each triplet of odorants we sampled $10^4$ combinations of RCI's—each RCI was randomly sampled from a Gaussian distribution with a mean equal to the experimental mean RCI and a standard error that was measured/estimated as described above. Each drawn triplet was then deterministically classified according to its type. From the classification of the $10^4$ repeats we calculated the probabilities of the triplet to be assigned to the four types of transitivity (Supplementary Table 2).

Although the ordinal approach has the advantage of being independent of the scale readout, it does not take into account preference magnitudes, and thus can over-estimate violations when preference levels are minor. In addition, it falls short in analyses of high preference ratios with low resolutions and high variability.

Therefore, we complemented this ranking approach with a parametric quantitative measure, based on the principles of the strict matching law, which links response to reward[48], and the product rule, which implies that preference magnitudes are exponentially additive[31,49]. Under the assumption that the strict matching law applies to our assay design (i.e., assuming that the end-point choice ratio is linear with the preference ratio), the product rule implies that the cyclic sum of end-point logits should be zero:

$$\log\frac{P_{(A/B)}}{P_{(B/A)}} + \log\frac{P_{(B/C)}}{P_{(C/B)}} + \log\frac{P_{(C/A)}}{P_{(A/C)}} = 0 \quad (1)$$

To avoid confusion with the traditional definition of the chemotaxis index, we use the term "Pair-Score" (PS) to describe the logit transformation of the assay end-point distribution $\left(\log\frac{P_{(A/B)}}{P_{(B/A)}}\right)$. Pair scores, probabilities (P), and relative chemotaxis indices (RCI) are transformable through the connections:

$$P_{(A/B)} = \frac{1 + RCI(A/B)}{2} \quad (2)$$

$$PS(A/B) = \log\left(\frac{P_{(A/B)}}{P_{(B/A)}}\right) = \log\left(\frac{P_{(A/B)}}{1 - P_{(A/B)}}\right) = \log\left(\frac{1 + RCI(A/B)}{1 - RCI(A/B)}\right) \quad (3)$$

Thus, $PS(A/B)$ represents a quantitative measure for how much option A is preferred over option B. The pair-score can receive values ranging from $−\infty$ to $+\infty$, and a pair-score of a given pair order is the negative of the same pair in reciprocal order, that is, $PS(A/B) = −PS(B/A)$.

We also define the cyclic summation of Pair Scores within a triple-set options as the Triple-Score (TS):

$$TS(A/B/C) = PS(A/B) + PS(B/C) + PS(C/A) \quad (4)$$

If preferences within a triple-set obey the transitivity rule, then the TS score will approximate zero. TS can also range from $−\infty$ to $+\infty$, and the magnitude by which a TS deviates from zero signifies the degree by which transitivity is violated. We used the same bootstrap method described above to also estimate the TS and its 95% confidence interval.

**Microfluidic-based system for generating smooth gradients**. Temporal chemical gradients were generated using computer-controlled syringe pumps that were used to flow the stimulus of choice and a diluting buffer into a small volume (50 μL) mixing chamber[17]. The content of the chamber was stirred using a magnetic bead and its homogenous output was flown through the nose of a worm constrained in a custom-made microfluidic device.

We used two syringes as the input to the mixing chamber, a "Buffer" syringe and a "Stimulus" syringe. The "Buffer" syringe contained CTM buffer and the "Stimulus" syringe contained the gradient stimulus, TT ($10^{-3}$) or diacetyl ($5 \times 10^{-5}$) volumetric dilution, in the CTM buffer. To verify the accuracy of the stimulus gradients we added to the "Stimulus" syringe a Rhodamine dye (0.2–1 μM). To allow accurate reading of neural activity, we added 10 mM levamisole (Sigma, CAS Number: 16595–80–5) to both syringes.

When imaging neural response to a stimulus in the background of a different stimulus, the background chemical, volumetric dilutions of TT ($6 \times 10^{-4}$) or diacetyl ($3 \times 10^{-5}$) were added to both syringes. Prior to imaging, worms were placed on empty NGM plates (w/o OP 50) for a short starvation period (10–40 min). Following a wash in CTM buffer, the worms were loaded into the microfluidic chamber.

**Calcium imaging of single neurons**. Imaging the AWB pair of neurons was performed using an Olympus IX-83 inverted microscope equipped with a Photo-metrics EMCCD camera and a × 40 magnification (0.95 NA) Olympus objective. A dual band filter (Chroma 59012) and a double-led illumination source (X-cite, Lumen Dynamics) were used to allow fast iterative imaging of the green and the red channels intermittently. Hardware was controlled using Micro-Manager[73]. Red dye (Rhodamine) was used to quantify the temporal concentrations of TT or DA. Movies, imaged at a frame rate of 1.4 Hz, were analyzed using MATLAB scripts developed in-house to extract neural activity.

**Calcium imaging of the entire chemosensory system**. Imaging was performed on a Nikon AR1 + fast-scanning confocal system controlled by the Nikon NIS-elements software. We used a water-immersed × 40 Nikon objective (1.15 NA) for imaging at a frame rate of 1.2 volumes/sec. Pinhole opening was one Airy units and z slice jumps were ~ 0.7 μm. Following the imaging, amphid neurons were detected and tracked offline based on the NLS-mCherry signal. Detection and tracking was done using a software package developed by Toyoshima et al.[74]. Following the tracking procedure, neurons were visually identified and calcium signals were retrieved using in-house Matlab scripts. For each neuron, mean calcium signal was calculated at a ratio of 0.7 from the radius of the nucleus to reduce possible readout of fluorescence signals from adjacent neurons. As activity of the AWB neurons was occasionally masked by the strong AWA activity, we also imaged the AWB neurons using a different strain in which GCaMP is expressed exclusively in the AWB neurons (Fig. 5i–j). As the two bilateral right-left symmetric neurons generally showed similar response dynamics, we averaged their activity profiles (Fig. 5e–l).

**Statistical analysis**. Data analysis was performed using custom Matlab scripts. All paired $t$ tests were performed between experiments performed at the same day to overcome between-days variability. As within-day variability was significantly lower than between-days variability (Supplementary Fig. 4), Chemotaxis indexes of all plates of the same condition within the same day were averaged and treated as a single measurement in all the performed $t$ tests.

**Reporting summary**. Further information on research design is available in the Nature Research Reporting Summary linked to this article.

## Data availability
The data used for the construction of the comprehensive pairwise preference matrix and the associated analyses (Fig. 1–3) are provided in Supplementary Tables 1–2. Any additional data information is available upon request.

## Code availability
All the code used in this study is available upon request.

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

## Acknowledgements

We thank Sergiu Hart, Maya Bar-Hillel, and Yonatan Lowenstein for fruitful discussions of this manuscript. Some strains were provided by the CGC, which is funded by the NIH Office of Research Infrastructure Programs (P40 OD010440). This study was funded by ERC (336803), ICORE, The American Federation for Aging Research, and ISF (1259/13, 1300/17) to AZ. COP postdoctoral fellowship is supported by the David-Herzog-Funds at Styrian Universities. EB, RR, and EI are supported by the Jerusalem Brain Center. AZ thanks the Joseph H. and Belle R. Braun Senior Lecture Chair fund.

## Author contributions

AZ conceived the study. SI performed all behavioral assays. RR and EI performed and analyzed the calcium-imaging experiments. COP and EB contributed valuable reagents. AZ, RR, and SI analyzed the results, interpreted the data, and wrote the paper.

## Additional information

**Competing interests:** The authors declare no competing interests.

