## [Peer Review File · Nature Communications]

Reviewers' Comments:

Reviewer #1:

Remarks to the Author:

This is a very interesting study but suffering from a number of major and minor problems that will need to be addressed by the authors in a major revision. I attached an annotated PDF with most of my comments and queries so I just list here what I regard the most important ones.

MAJOR

1 Title. This creates the impression that the findings are from a human or at least primate study so it needs to be changed to "Irrational behavior in *C. elegans* arises due to...". It is important to tell the reader what the experimental animal was as in my view the potential generality of the otherwise very interesting findings is questionable (also see under 2.).

2. No statistical analysis is provided to support the notion that important decisions are made at the level of individual sensory cells. The authors refer to 'significant' and 'marked' or 'mild' effects but without providing any statistics. In the absence of strong statistical support for these statements, the whole idea of how the behaviorally observed asymmetric effects originate in specific sensory neurons falls apart. But even if this hypothesis can be supported by statistics, I have doubts about the potential generality of findings in *C. elegans*, as the sensory systems of vertebrates and even invertebrates other than *C. elegans* might operate in a different way with sensory perception integrated more centrally.

3. I disagree with the notion that because *C. elegans* are isogenic and grow under the same conditions they are devoid of all the confounding effects mentioned in the manuscript. This is not necessarily the case and there may be some differences in individual worms in past experience and current state.

4. The manuscript doesn't have a clear structure and a particular issue is the absence of a proper discussion in the context of findings in other systems. As it stands at the moment, the brief paragraph that can be regarded as an attempt at discussion is just a recapitulation of the findings from the experiments.

Reviewer #2:

Remarks to the Author:

In this manuscript, Iwanir et al. demonstrate that *C. elegans* can display "irrational" behavior in terms of its chemosensory preferences. By testing different chemosensory stimuli in a behavioral choice assay, they show that in a few cases, chemosensory preferences among groups of odorants are intransitive. They show that this irrationality also exists when worms are presented with some odorants in a background of a different odorant. Finally, they image head chemosensory neurons and show that different odorants elicit different sensory neuron responses, which may contribute to the observed behavioral intransitivity. The fact that chemosensory preferences are not always "rational" is interesting and perhaps unexpected. However, I have a number of major concerns about the significance of these results and the authors' interpretation of their results. These issues are detailed

below.

1) Irrationality, as measured by intransitivity among groups of three odor-preference assays, is extremely rare. The authors observed intransitivity in only 0.5% of the 3,276 triple preference sets tested. Thus, *C. elegans* behaves rationally in the vast majority of tested cases. It doesn't make sense to me to use *C. elegans* as a model for understanding irrational behavior in other organisms when this irrational behavior seems to be an exception to the rule. In addition, this entire paper is framed as a *C. elegans* model for understanding irrational behavior in higher organisms, including humans. I'm highly skeptical that these experiments will tell us anything about irrational behavior in other organisms.

2) On the flip side of this, I am somewhat surprised that intransitivity is not more common when it comes to olfaction, given the organization of the nematode olfactory system. Nematodes have only a few olfactory neurons, each of which expresses many odorant receptors. It seems that this organization could easily lend itself to intransitivity. For example, in the case of TT, DA, and IA, is it possible that intransitivity arises because TT and DA activate overlapping subsets of neurons? If TT and DA both strongly activate AWA, then information from AWA might be less useful in a preference test, and relative preference may instead be determined by which other neurons are also activated. The authors state that intransitivity arises from a scenario involving "intricate asymmetric activities of individual sensory neurons," but aren't much simpler explanations also possible?

I also think it's odd that the panel in Fig. 1B includes odorants as well as tastants. The main olfactory neurons are different from the main gustatory neurons, so comparing discrimination among odorants to discrimination among odorants vs. tastants doesn't seem like a fair comparison. I wonder how the data would look if the panel consisted entirely of odorants. With a larger odorant panel, would intransitivity be more common?

3) Related to that point, almost all of the cases of intransitivity involved the same three odorants. I would be more convinced about the ethological significance of the intransitivity cases if they involved more combinations of odorants. Also, is it possible that there's something particular about the chemical properties of these odorants that contributes to intransitivity?

Other comments:

1. A big concern in terms of the experimental setup is that the data seem very underpowered. All of the chemotaxis data seems to be based on only 2-3 replicates, with 2-3 plates per experiment. Given the day-to-day variability that often arises with *C. elegans* chemotaxis assays, I'm skeptical that 2-3 replicates are sufficient to make the data robust. I realize that doing more replicates on the entire matrix shown in Figure 1B is not realistic. However, I think it's important for the authors to confirm their results with selected combinations with a much larger sample size, collected over multiple days. I do see data for one combination in Figure S3, but this seems to be the only combination confirmed with a larger sample size.

If the authors were simply looking at which stimuli elicited a response, sample size would be much less of a concern. However, the authors' conclusions are much more specific. In particular, some of the results shown in Figure 1C seem to be based on very quantitative comparisons of chemotaxis indices across assays (SST, MST, and WST). It is important to confirm that these results are reproducible with a larger sample size. On a related note, the exact sample sizes for each experiment should be specified in the figure legends.

2. In Figure 4, wouldn't it make more sense to look at odorant combinations that were not already

shown to be intransitive? It would be more interesting to find odorants that don't show intransitivity but violate IIA and rationality.

3. Although the calcium imaging data demonstrate that the tested odorants activate a number of different sensory neurons, this provides very little insight into how intransitivity arises. A major experiment that's lacking is any kind of neuronal manipulation (genetic ablation or silencing, optogenetics, etc.). I see in the methods that a number of interesting mutants are listed (e.g., odr-10, odr-7) but I don't see the data from these strains anywhere. It would obviously be really informative in terms of mechanism to find a neuron that is not required for response to a particular odorant but that affects transitivity specifically. Also, have the authors tried imaging responses to odorants that form a transitive set? Without either comparative imaging using a different set of odorants, or neuronal manipulation, it's really hard to know what to make of the imaging data.

Reviewer #3:

Remarks to the Author:

In this manuscript the authors introduce a *C. elegans* chemotaxis model for studying decision making behaviors, and using it show that worms' innate behavior is not rational, violating transitivity, independence of irrelevant alternatives, and regularity.

They demonstrate that the rationality violations are due to asymmetric modulatory effects at the level of sensory neurons: the presence of a uniform odorant of one type can bias or even reverse the animal's preference to another odorant, and in an asymmetric manner.

The results provide an interesting dissection of cross modulation in chemosensation. However I have a number of serious concerns about the paper.

- The finding that one uniform odorant can affect the chemotaxis to another odorant is not really new. Bargmann and Horvitz (1993) reported cross saturation between different odorant species; high concentrations of certain odorants eliminated chemotaxis responses to certain other odorants. The same paper also established that a single odorant may be attractive or repulsive depending on the concentration. Indeed, odorants will modulate their own attractiveness in the sense that adding a background concentration can switch the direction of the animals' preference. Given this context, the results of the manuscript are not very surprising.

- The manuscript is couched in a language of explaining rationality in human decision making, for which this manuscript has very little relevance.

As a result, the context for the present experiments is not properly introduced or discussed. The authors should discuss the literature on behavioral choice and optimization in general and in *C. elegans* in particular.

- The authors summary of their results as "...suggesting that asymmetric representations may provide a simple explanation for irrational behavior" deserves much more explanation and discussion. Are the authors claiming that asymmetric representations explain human irrational behavior? If so, how?

Point by point reply to reviewers' comments

We wish to thank the reviewers whose comments aided us to significantly improve the manuscript. Below, please find our detailed point-by-point response in which we have fully addressed all of the concerns. Specifically, we have now added new data of mutant strains, defective in individual sensory neurons, that together with the calcium imaging data, allowed us to propose a simple mechanism for the observed asymmetry and the consequent irrational behavior (a mechanism also proposed by reviewer #2). Furthermore, we have now added a thorough statistical analysis of the comprehensive choice-assay screen that provides accurate characterization of the results. These experiments along with the extensive analyses yielded two new figures to the main text and two new supplementary figures. In addition, we have restructured the manuscript, provided a detailed description of the various analyses in the methods section, and added a detailed introduction and discussion sections.

Reviewer #1 (Remarks to the Author):

This is a very interesting study but suffering from a number of major and minor problems that will need to be addressed by the authors in a major revision. I attached an annotated PDF with most of my comments and queries so I just list here what I regard the most important ones.

MAJOR

1 Title. This creates the impression that the findings are from a human or at least primate study so it needs to be changed to "Irrational behavior in *C. elegans* arises due to...". It is important to tell the reader what the experimental animal was as in my view the potential generality of the otherwise very interesting findings is questionable (also see under 2.).

We have now changed the title to:

"Irrational behavior in *C. elegans* arises due to asymmetric modulatory effects between options"

2. No statistical analysis is provided to support the notion that important decisions are made at the level of individual sensory cells. The authors refer to 'significant' and 'marked' or 'mild' effects but without providing any statistics. In the absence of strong statistical support for these statements, the whole idea of how the behaviorally observed asymmetric effects originate in specific sensory neurons falls apart.

We have now included a detailed statistical analysis that further underscores the roles of individual neurons in the differential modulated coding. The new Supplementary figure 6 depicts the neurons that significantly responded in each of the tested conditions (of the calcium imaging in fig. 5). Specifically, we used the Wilcoxon signed-rank test followed by Benjamini-Hochberg FDR correction for multiple comparisons to extract those neurons with significant responses. We have now added to the results that the extracted neurons with significantly modulated activity is based on statistical tests, and refer to suppl. fig 6, where its legend describes the statistical analysis.

On page 6:

"We identified several chemosensory neuron types whose mean activity significantly changed after the gradient onset (Supplementary Fig. 6)."

Supplementary figure 6:

“Supplementary figure 6. Statistical analysis of activated sensory neurons. (a-d) Mean fold change fluorescence in the first 60 seconds after gradient onset (y-axis) is plotted as a function of mean fold change fluorescence in the 50 seconds before gradient start (x-axis) in four experimental conditions: TT/DA gradients without the reciprocal stimulus in the background (a,b) and with the reciprocal stimulus in the background (c,d). Plotted are the mean values of each of the 11 neuron pair types that were imaged in Figure 5 a-d. The 11 neurons in each condition were subjected to Wilcoxon signed-rank test followed by Benjamini-Hochberg FDR correction. (*) marks neuron types whose mean fold-change significantly varied between the two time periods ($p < 0.05$). Light sensitive neurons (ASK, ADL, ASH) showed significant reduction in activity throughout the recording due to light adaptation regardless of the gradient. These analyses are based on the *osm-6::GCaMP* reporter strain where all neurons were measured simultaneously. The AWB neurons were measured using a separate strain with AWB-exclusive expression (as shown in Figure 5).”

Furthermore, we now provide statistical analysis showing that activities of AWA and AWC neurons were differentially activated between the different conditions (in the different backgrounds). In the legend of Fig 5:

“Mean AWA activity in the first 150 seconds of the gradient was significantly weaker for both DA and TT in their reciprocal background presence compared to their respective null background conditions (blue vs red curves in e,f, ranksum test, $p = 0.016$ and $p = 0.006$, respectively). Mean AWC activity in response to a TT gradient was significantly elevated during gradient application in the presence of a DA background but not in its absence (blue vs red

curve in g, ranksum test, $p=0.0015$). In e-l, the number of neurons imaged with and without background (red and blue, respectively) is: {e:10,20}; {f:17,18}; {g:12,19}; {h:11,15}; {i:7,5}; {j:5,8}; {k:14,20}; {l:13,19}.”

But even if this hypothesis can be supported by statistics, I have doubts about the potential generality of findings in *C. elegans*, as the sensory systems of vertebrates and even invertebrates other than *C. elegans* might operate in a different way with sensory perception integrated more centrally.

Thank you for pointing this out. We now clarify in the discussion what can be drawn from our studies in *C. elegans*. It is true that the sensory system in *C. elegans* operates differently than in vertebrates and other invertebrates. However, the basic principle that we elucidated, where options can asymmetrically modulate the way they are sensed (perceived), may be equally relevant for vertebrates, and even humans. Studying *C. elegans* worms allowed us to unequivocally show this asymmetric modulation with unparalleled cellular resolution.

Furthermore, we postulate that if a single neuron can switch animals' behavior from attraction to retraction (which is intriguing by itself), then similar processes may take place in higher brain systems, if not at the single neuron level, possibly at the circuit/brain area level.

We now discuss the generality of our findings, providing analogous examples in humans and the possible relevance to higher brain systems (p. 11):

“Interestingly, analogous trans-modulatory effects that lead to irrational behavior were observed in humans as well as in other vertebrate and invertebrate animals^{15,25,27,59}. These are known as the asymmetric dominance effect, or the decoy effect, a form of context-dependent choice in which the probability of choosing one of two options is impacted by the introduction of a third weaker option⁶⁰. According to this effect, adding to a set of options, A and B, a third option C that is asymmetrically dominated by option A (but not by option B), will increase the preference of choosing A over B. In humans, this effect had been extensively documented in various fields, including marketing and consumption⁶¹, policy decisions⁶², and partner preferences⁶³.”

*Obviously, mammalian brains are organized completely differently than the *C. elegans* neural network, and irrationality may arise due to complex integrated dynamics within different brain regions, rather than in individual perceptual neurons. Yet, it is possible that the same principles of asymmetric cross- or trans-modulatory effects between options may serve as the underlying basis that leads to irrational behavior.”*

3. I disagree with the notion that because *C. elegans* are isogenic and grow under the same conditions they are devoid of all the confounding effects mentioned in the manuscript. This is not necessarily the case and there may be some differences in individual worms in past experience and current state.

We agree and have now changed accordingly. We acknowledge that *C. elegans* worms have inherent variability and that being isogenic and growing under the exact same conditions, merely reduces this variability.

We now added in the introduction, pp. 2- 3:

"In addition, while hermaphroditic C. elegans worms maintain variability at the individual level³¹⁻³³, the fact that they are isogenic, and grown under the exact same conditions, reduces variability between individuals."

4. The manuscript doesn't have a clear structure and a particular issue is the absence of a proper discussion in the context of findings in other systems. As it stands at the moment, the brief paragraph that can be regarded as an attempt at discussion is just a recapitulation of the findings from the experiments.

We have now substantially revised the manuscript by adding extended introduction and discussion sections. As we detailed in the reply to issue #3 above, we also placed our findings in the context of human brain and behavior (p. 11).

We append below the comments raised in an attached PDF file, and which were not mentioned above.

Abstract:

"Humans, like animals..." the other way ?

Thank you. We have now changed the order accordingly:

"Animals, like humans, ..."

Bibliography:

Bargmann, C. I. Chemosensation in C. elegans. WormBook (2006). - Is this Ref OK?

We have now revised the reference:

Bargmann, C. I. Chemosensation in C. elegans. WormBook, doi:10.1895/wormbook.1.123.1 (2006).

Figure 2 - ???

Figure 2 is now Figure 4 in the revised manuscript.

The mild asymmetry between IA and TT is now apparent as we did not find that IA significantly affects TT while TT affects IA (see fig. 4c). We therefore eliminated one of the arrows to emphasize the mild asymmetry. We use the term mild in comparison to the strong asymmetry observed between TT and DA.

Reviewer #2 (Remarks to the Author):

In this manuscript, Iwanir et al. demonstrate that C. elegans can display "irrational" behavior in terms of its chemosensory preferences. By testing different chemosensory stimuli in a behavioral choice assay, they show that in a few cases, chemosensory preferences among groups of odorants are intransitive. They show that this irrationality also exists when worms are presented with some odorants in a background of a different odorant. Finally, they image head chemosensory neurons and show that different odorants elicit different sensory neuron responses, which may contribute to the observed behavioral intransitivity. The fact that chemosensory preferences are not always "rational" is interesting and perhaps unexpected. However, I have a number of major concerns about the significance of these results and the authors' interpretation of their results. These issues are detailed below.

1) Irrationality, as measured by intransitivity among groups of three odor-preference assays, is extremely rare. The authors observed intransitivity in only 0.5% of the 3,276 triple preference sets tested. Thus, C. elegans behaves rationally in the vast majority of tested cases.

It doesn't make sense to me to use *C. elegans* as a model for understanding irrational behavior in other organisms when this irrational behavior seems to be an exception to the rule.

Thank you, this issue indeed requires a clarification. We note that the fraction of intransitive sets greatly depends on the specific stimuli we chose for the analyses. For example, had we chosen only same-type of stimuli, sensed by the same neurons, we would probably obtain a much higher % of intransitivity (as can also be seen in the newly added fig. 3). Moreover, our statistical power did not allow us to conclusively classify many of the sets into one of the in/transitive types. Thus, we have considerably underestimated the fraction of intransitive sets.

In an effort to overcome this shortage and to provide a better estimate of the fraction of intransitive sets, we first classified each triple set according to its most probable type of intransitivity (or transitivity) without assuming any prior that all triplets are transitive (and in contrast to the approach that was used in the original analysis). The new analysis showed that full intransitivity occurs in ~7% of all triple sets (now appears in the revised version as the new fig 3).

A more conservative approach which we also applied in our new analysis focused on a subset of the triplets that had the significance power to be classified into one of the in/transitive types (either transitive or violating SST/MST/WST). These included 351 triplets of which 4% of the sets were fully intransitive, and a total of 16% of the sets violated any type of intransitivity (Fig. 3b).

We now added the detailed statistical analysis to the results which also appears in the new fig. 3. In the results section, on page 4:

*“Overall, our large-scale behavioral screen consisted of 3,276 triple sets; however, many of the choice comparisons were based on a small number of experimental repeats (Fig. 1). To overcome this, we constructed a noise model, which allowed us to provide a more accurate classification of the various triple sets (see methods and **Supplementary Fig. 4**). We found that 351 triplets could be classified with statistical significance (Fig. 3b), of which 243 triplets were heterogeneous, (i.e., consisted of three different types of options, Fig. 3c). Interestingly, only a small fraction (3%) of these 243 heterogeneous triplets were fully intransitive. Furthermore, violations of transitivity were more frequent in triplets consisted of odorants only, rather than in triplets consisted of both odorants and tastants (Fig. 3d-e, $p < 0.01$, χ^2 test).”*

Having performed this analysis, we believe that these numbers need to be taken with caution as they heavily depend on the specific experimental details (e.g., stimuli and statistical power). In the results we now added the following, pp. 4-5:

“Notably, while our analyses indicate that the fraction of total intransitive cases is ~10%-15%, this percentage greatly depends on the specific set of stimuli used to construct the matrix, as well as on the statistical power for inferring in(transitivity).”

Furthermore, we would like to emphasize that we generated the comprehensive choice-assay matrix primarily as a screen to identify conditions which violate transitivity. These conditions allowed us to subsequently dissect the possible causes that lead to irrationality. In such case, even if irrationality is uncommon, deciphering the conditions that lead to it provides powerful means to study the underlying mechanism.

In the discussion, we state that the matrix was used merely to identify intransitive conditions.
Page 9:

"First, we constructed a comprehensive pairwise preferences matrix (Fig. 1) which allowed us to uncover sets of options that consistently violated the transitivity rule (Fig. 2)."

In addition, this entire paper is framed as a *C. elegans* model for understanding irrational behavior in higher organisms, including humans. I'm highly skeptical that these experiments will tell us anything about irrational behavior in other organisms.

We agree that due to the vast differences between mammalian brains and the *C. elegans* neural network, irrationality in higher brain systems may involve various brain regions with complex dynamics between them. However, we speculate that the same ideas of asymmetric modulatory effects between options may underlie irrationality in various brain systems. In particular, the famous *asymmetric dominance effect*, which is extensively studied in the context of human decision making can be viewed as analogous to the asymmetric trans modulation that we observe in *C. elegans*. Furthermore, the use of *C. elegans* as the model system allowed us to propose a mechanistic understanding for this irrational behavior at an unprecedented single-neuron level, thus, an equivalent mechanism may underlie irrationality in higher brain systems at the cellular/circuitry level.

We now explicitly discuss this on page 11:

"Interestingly, analogous trans-modulatory effects that lead to irrational behavior were observed in humans as well as in other vertebrate and invertebrate animals^{15,25,27,59}. These are known as the asymmetric dominance effect, or the decoy effect, a form of context-dependent choice in which the probability of choosing one of two options is impacted by the introduction of a third weaker option⁶⁰. According to this effect, adding to a set of options, A and B, a third option C that is asymmetrically dominated by option A (but not by option B), will increase the preference of choosing A over B. In humans, this effect had been extensively documented in various fields, including marketing and consumption⁶¹, policy decisions⁶², and partner preferences⁶³.

*Obviously, mammalian brains are organized completely differently than the *C. elegans* neural network, and irrationality may arise due to complex integrated dynamics within different brain regions, rather than in individual perceptual neurons. Yet, it is possible that the same principles of asymmetric cross- or trans-modulatory effects between options may serve as the underlying basis that leads to irrational behavior."*

2) On the flip side of this, I am somewhat surprised that intransitivity is not more common when it comes to olfaction, given the organization of the nematode olfactory system. Nematodes have only a few olfactory neurons, each of which expresses many odorant receptors. It seems that this organization could easily lend itself to intransitivity. For example, in the case of TT, DA, and IA, is it possible that intransitivity arises because TT and DA activate overlapping subsets of neurons? If TT and DA both strongly activate AWA, then information from AWA might be less useful in a preference test, and relative preference may instead be determined by which other neurons are also activated. The authors state that intransitivity arises from a scenario involving "intricate asymmetric activities of individual sensory neurons," but aren't much simpler explanations also possible?

Thank you, this is indeed a possibility that we have now thoroughly studied by adding new experiments using mutant animals. First, we corroborated that when considering olfactory cues only, the percent of intransitive sets is indeed higher (6% when considering mixed triplets and 17% when considering volatiles only, fig. 3d-e).

We have now performed new experiments using mutants defective either in individual sensory neurons or in the DA receptor ODR-10. These experiments indeed support the idea that there is a sensory overlap between DA and TT on the AWA neurons which may lead to the observed intransitivity.

The results, together with the suggested model, appear in the new figure 6, and detailed on pages 8-8 under the section: 'Mutant analysis reveals that cross-modulatory effects at a single neuron type, AWA, may mediate irrational behavior'.

The following, copy-pasted from the results section on page 8, summarizes the suggested model:

"The neuroimaging analyses (Fig. 5), together with the chemotaxis behavior obtained by mutant strains (Fig. 6 a-d), offer a model where the DA-mediated TT repulsion may lead to asymmetric modulation and irrationality (Fig. 6e): In the absence of DA, TT is sensed by a, yet to be identified, receptor on the AWA neurons, whose strong activation mediates attraction (Fig. 5e, Fig. 6a,c). Additional sensory neurons, including AWC, mediate weak repulsion from TT. In the presence of AWA, this repulsion has no effect on chemotaxis preferences (Fig. 6a,c,e-left panel). When DA is present, it is sensed by the AWA neurons through the ODR-10 GPCR (Fig. 6d), leading to a decreased activity in AWA neurons in response to TT (Fig. 5e). This leads to the observed AWA-mediated repulsion (Fig. 6c). In addition, AWC and other sensory neurons mediate basal TT repulsion, independent of the DA background (Fig. 6c). Together, the combined activity of these neurons shifts the worm preference from strong attraction to repulsion (Fig. 6e-right panel)."

Figure 6

And in the discussion (page 10):

“Thus, the behavioral switch, and concomitantly the irrational behavior, may largely result from input integration occurring within a single type of neurons, AWA, where each input (stimulus) is sensed by its own cognate receptor...”

“Stimuli converging onto the same sensory neurons may also underlie the other instances of intransitive triplets that we discovered throughout the comprehensive pairwise choice assays (Figs. 1, 3). In fact, we found that intransitivity is significantly more common when considering same-modality stimuli (e.g., odorants only, Fig. 3c-d), which are sensed by the same sensory neurons. Together, modulatory effects of stimuli, that change neural activity and behavioral outputs, may eventually lead to irrational decision making in C. elegans.”

I also think it's odd that the panel in Fig. 1B includes odorants as well as tastants. The main olfactory neurons are different from the main gustatory neurons, so comparing discrimination

among odorants to discrimination among odorants vs. tastants doesn't seem like a fair comparison. I wonder how the data would look if the panel consisted entirely of odorants. With a larger odorant panel, would intransitivity be more common?

We have now analyzed odorants separately from a mixture of odorants and tastants. These results now appear in a newly added Figure 3. Indeed, we find that when considering odorants only, a statistically significant higher fraction of the sets violates transitivity (of any type) when compared to sets comprising of a mixture of odorants and tastants (see also our response from previous comment). This, together with the data obtained from mutant analysis and the Ca imaging experiments, supports the model suggested above that intransitivity may arise because stimuli are sensed by overlapping subsets of neurons. We now added this suggested model in Figure 6e.

3) Related to that point, almost all of the cases of intransitivity involved the same three odorants. I would be more convinced about the ethological significance of the intransitivity cases if they involved more combinations of odorants. Also, is it possible that there's something particular about the chemical properties of these odorants that contributes to intransitivity?

Indeed, many subsets that violate transitivity involved the same three odorants, and on which we decided to focus in this manuscript. However, other stimuli equally appear in intransitive sets. In fact, we extracted from the supplementary table 2 all fully intransitive heterogeneous sets (8 intransitive sets out of 243 triplets in total, Fig 3c), and found that they consist of the vast majority of the stimuli used in this study:

- DA-2/NH4Ac-2E+1/PD-4
- DA-4/PD-2/Py-2
- DA-4/PD-2/Py-3
- DA-4/PD-2/TT-2
- DA-4/PD-3/Py-3
- IA-1/NaCl-5E-0/TT-2
- IA-1/PD-4/TT-2
- IA-2/NaCl-5E-0/TT-2

As we noted in the above response to comment #1, the fraction and the type of stimuli leading to intransitivity will greatly depend on the specific chosen triplets, their concentrations, and the number of repeats that allow to draw statistical meaning.

Regarding the ethological significance: we used stimuli known to attract *C. elegans* worms. Moreover, some of these stimuli had been shown to be secreted from bacteria upon which the worms feed (Choi, Yoon et al. 2016, Worthy, Haynes et al. 2018). It is conceivable, however, that the concentrations used in our assays do not necessarily reflect the concentrations found in ecological niches.

Notably, our main goal in constructing the choice-preference matrix was to screen for conditions that lead to genuine intransitive behaviors. We then focused on intriguing instances for in depth analyses that allowed us to infer the mechanisms that underlie such irrational behavior.

We have now added all these considerations to the discussion, pages 9-10:

" To construct the comprehensive choice matrix (Fig. 1), we used stimuli that are known to attract C. elegans worms²⁹. Moreover, some of these stimuli had been shown to be secreted by bacteria, upon which the worms feed^{55,56}. While the concentrations used in our assays may not reflect the typical concentrations found in the natural ecological niches, they still served as potent attractors; thus, the irrationalities reported herein are found in the context of the worms' innate chemotaxis behavior. This crucial feature allowed us to overcome the requirement for initial training procedures that often bias rational behavior studies."

We are not aware of something particular about the chemical properties of DA and TT, except that the DA receptor, ODR-10, is found exclusively on AWA neurons, and that there is a different unknown receptor for TT, presumably expresses on AWA neurons as well (based on mutants' analysis, fig. 6). Their combined effect on the sensory neurons dictates the behavioral outputs with no apparent relation to their chemical properties.

Other comments:

1. A big concern in terms of the experimental setup is that the data seem very underpowered. All of the chemotaxis data seems to be based on only 2-3 replicates, with 2-3 plates per experiment. Given the day-to-day variability that often arises with C. elegans chemotaxis assays, I'm skeptical that 2-3 replicates are sufficient to make the data robust. I realize that doing more replicates on the entire matrix shown in Figure 1B is not realistic. However, I think it's important for the authors to confirm their results with selected combinations with a much larger sample size, collected over multiple days. I do see data for one combination in Figure S3, but this seems to be the only combination confirmed with a larger sample size. If the authors were simply looking at which stimuli elicited a response, sample size would be much less of a concern. However, the authors conclusions are much more specific. In particular, some of the results shown in Figure 1C seem to be based on very quantitative comparisons of chemotaxis indices across assays (SST, MST, and WST). It is important to confirm that these results are reproducible with a larger sample size. On a related note, the exact sample sizes for each experiment should be specified in the figure legends.

Indeed, as we strived to construct a comprehensive behavioral screen, many sets were under sampled, thus precluding from assigning them with accurate statistical significance.

To address this issue and to provide a better estimate of the standard error within our data, particularly for the pairs with the small size of experimental repeats, we have now constructed a noise model (appears as a new supplementary fig. 4). For this, we compiled a list of all pairwise choice assays from the big data set (46 in total) for which we performed at least 5 independent experimental repeats (that is, performed over 5 different days) where each experimental repeat consisted of 2-4 replica plates. Importantly, this list of choice assays included a variety of representative stimulants including odorants, tastants, ctrl stimuli, which formed a range of weak, intermediate and strong chemoattractants.

Analysis of the experimental noise indicated that the major noise in the system is indeed the day-to-day variability, rather than the within day variability (Suppl. Fig 4a). We calculated the linear regression between the various RCI values and the day-to-day standard deviation ($R=0.63$, $p<10^{-5}$, suppl. Fig 4b) and used this regression to approximate the standard deviation in experiments for which we had less than five experimental repeats.

This analysis is now detailed in the methods section, page 15:

“Estimation of the standard deviation

When constructing the binary preference matrix, most binary choice preferences were estimated based on 1-4 experimental between-day repeats. In order to better estimate the variance in worm's preferences (RCI, BCI), we chose 46 different pairs of stimuli (consisted of odorants, tastants, the buffer null-option, and which span various magnitudes of CI's), and assayed them on at least 5 different days with ~4 plate replicas on each of the different days. Analysis of these experimental repeats revealed that variance between days was significantly higher than the variance within the same day experiments (Supplementary Fig. 4a). We therefore used the between-days variance to assess the CI variance. Additionally, we found a strong negative correlation between the CI absolute value and the between-days standard deviation ($R=-0.63$, $p<10^{-5}$, Supplementary Fig. 4b). Consequently, we used this linear regression to estimate the day-to-day standard deviation for pairs of stimuli for which we had less than 5 experimental repeats on different days.”

Thus, the large-scale screen was used to extract possible intransitive sets, for which we then corroborated one of them in-depth. These large-scale data were also used, with the aid of the noise model we constructed, to demonstrate the overall magnitude of intransitivity in our data set (fig. 3).

We also modified the text and the legend in fig. 2a (previously figure 1c) to state that these are putative sets which violate different types of transitivity, and we also provide the number of experimental repeats in the figure legend. For the set that we chose to focus on in this study, we corroborated its significant full-intransitivity via 14 experimental repeats (suppl. fig. 2).

We now state that we found putative sets that violate transitivity (page 4):

“Using our data set, we were able to detect putative triplets that violated all the different types of transitivity (Fig. 2a, Supplementary table 2).”

And modified the legend of figure 2:

“Levels of violations of stochastic transitivity (top, schematic; bottom, examples for putative sets)... For each pairwise comparison, number of experimental repeats is provided in the order of {left pair, right pair, top pair}”

2. In Figure 4, wouldn't it make more sense to look at odorant combinations that were not already shown to be intransitive? It would be more interesting to find odorants that don't show intransitivity but violate IIA and rationality.

By using the same odorant combination, we wished to demonstrate how asymmetric effects between options that lead to intransitivity can also lead to violations of other rationality paradigms. In a sense, we wished to use the findings regarding the mutual effects gained thus far to educationally 'engineer' conditions that also violate IIA and regularity. For this, we used our findings that TT asymmetrically trans-modulates IA and DA (Fig. 4 in the revised manuscript) to change the preference between these two choices in a way that violates regularity and IIA.

We now explicitly state this in the discussion (page 10):

“While focusing on transitivity, we speculated that additional paradigms of irrational behavior may be violated due to differential modulatory effects between options. Following the finding of modulatory effects that TT exerts on IA (but not on DA, Fig. 4b-c), we were able to design experimental conditions that will exploit this asymmetry to demonstrate violations of IIA and regularity (Fig. 7). In this case, trans-modulatory effects, rather than cross-modulatory effects, lead to violations of these paradigms, where option C (TT) modulates the preference to option A (DA) differently than it modulates the preference to option B (IA) (Fig. 8b).”

3. Although the calcium imaging data demonstrate that the tested odorants activate a number of different sensory neurons, this provides very little insight into how intransitivity arises. A major experiment that's lacking is any kind of neuronal manipulation (genetic ablation or silencing, optogenetics, etc.). I see in the methods that a number of interesting mutants are listed (e.g., odr-10, odr-7) but I don't see the data from these strains anywhere. It would obviously be really informative in terms of mechanism to find a neuron that is not required for response to a particular odorant but that affects transitivity specifically.

Also, have the authors tried imaging responses to odorants that form a transitive set? Without either comparative imaging using a different set of odorants, or neuronal manipulation, it's really hard to know what to make of the imaging data.

Thank you. We followed this suggestion and performed new experiments using mutants in which the AWC and AWA neurons are defective. The results of these experiments are provided in the newly-added figure 6. As detailed in our reply to the above major comment no. 2, these new data, complemented the calcium imaging results, allowed us to propose a parsimonious mechanism for the observed intransitivity. Briefly, these data indicate that the DA-mediated TT repulsion occurs via the ODR-10 receptor on the AWA neurons, while the basal TT repulsion is mediated by the AWC neurons, and possibly additional neurons. When DA and TT appear together, the overall effect is repulsion. An extended explanation is found in our reply above and in the text describing figure 6. Figure 6e provides the proposed mechanism.

Reviewer #3 (Remarks to the Author):

In this manuscript the authors introduce a *C. elegans* chemotaxis model for studying decision making behaviors, and using it show that worms' innate behavior is not rational, violating transitivity, independence of irrelevant alternatives, and regularity.

They demonstrate that the rationality violations are due to asymmetric modulatory effects at the level of sensory neurons: the presence of a uniform odorant of one type can bias or even reverse the animal's preference to another odorant, and in an asymmetric manner.

The results provide an interesting dissection of cross modulation in chemosensation. However I have a number of serious concerns about the paper.

- The finding that one uniform odorant can affect the chemotaxis to another odorant is not really new. Bargmann and Horvitz (1993) reported cross saturation between different odorant species; high concentrations of certain odorants eliminated chemotaxis responses to certain other odorants. The same paper also established that a single odorant may be attractive or repulsive depending on the concentration. Indeed, odorants will modulate their own attractiveness in the sense that adding a background concentration can switch the direction of the animals' preference. Given this context, the results of the manuscript are not very surprising.

Indeed, previous studies showed that certain odorants abolish chemotaxis to other odorants, presumably via cross saturation, and that increasing concentrations of an odorant may switch behavior from attraction to repulsion. In our work, we showed that the addition of a second odorant not only eliminates attraction, but actually switches behavior from attraction to repulsion. Moreover, our newly-added data shows that this behavioral switch occurs within the AWA neurons through ODR-10-mediated DA sensation. This is a new mechanism for repulsion that adds to the existing known one, in which aversion to high concentrations is mediated by nociceptor neurons such as AWB and ASH (Kazushi et al. Nat Comm. 2012).

We now discuss the known cross saturation between two stimuli and the possible behavioral switch due to high odor concentrations, and also specify the novelty in our findings (page 10):

*"It was previously shown that the presence of a background stimulus can interfere with chemotaxis towards another stimulus⁴⁵. Here, we found that the presence of an attractive background stimulus not only affected attraction, but actually reversed the preference from attraction to repulsion. *C. elegans* worms may switch preferences from attraction to repulsion when a stimulus concentration increases beyond a certain threshold^{42,45,57}. These repulsive responses often trigger the nociceptive neurons AWB and ASH⁴²; however, our calcium imaging data indicate that activity of these neurons is not modulated between the reciprocal conditions. Instead, we found that AWA, and possibly AWC, are the major players contributing to the DA-mediated TT repulsion. Moreover, the data suggest that TT and DA are sensed by two different receptors on the AWA neurons (Fig. 6). Thus, the behavioral switch, and concomitantly the irrational behavior, may largely result from input integration occurring within a single type of neurons, AWA, where each input (stimulus) is sensed by its own cognate receptor. Interestingly, such a dual role for a single neuron type was suggested for the AWC neurons, which may mediate both attraction and repulsion from butanone via differential synaptic signaling⁵⁸. "*

Furthermore, to our knowledge, we are the first to place such cross-modulatory effects in the context of rationality, providing a mechanistic cellular-resolution view of how such modulations may actually lead to irrational behaviors.

Thus, we also added on page 10:

*"Together, modulatory effects of stimuli, that change neural activity and behavioral outputs, may eventually lead to irrational decision making in *C. elegans*. "*

- The manuscript is couched in a language of explaining rationality in human decision making, for which this manuscript has very little relevance. As a result, the context for the present experiments is no properly introduced or discussed. The authors should discuss the literature on behavioral choice and optimization in general and in *C. elegans* in particular.

Thank you for this important point that helped us to clarify the manuscript. In the revised version, we made substantial changes to properly introduce and discuss the *C. elegans* experimental system. Moreover, we have now added new experiments with mutant worms that elucidate the mechanism for irrationality in *C. elegans*. We then discuss these *C. elegans*-based findings in the broader context of the rational decision-making paradigms and in light of what is known from human and animal model systems.

All these changes appear throughout the new extended sections of the introduction, results and discussion.

For example, we provided a detailed description of *C. elegans* as a model animal on page 3: "...while hermaphroditic *C. elegans* worms maintain variability at the individual level³¹⁻³³, the fact that they are isogenic, and grown under the exact same conditions, reduces variability between individuals.

C. elegans worms have a compact fully-mapped nervous system, allowing to study activity dynamics of individual neurons³⁴. Its chemosensory system can sense and distinguish among a rich set of positive and negative chemical cues, and translate these signals into acute and lasting behavioral outputs such as attraction or repulsion^{29,35}. In addition, a myriad of experimental tools, including functional imaging of multiple neurons and neuron-specific genetic manipulations are available³⁶⁻³⁸, offering the unique opportunity to delineate the circuit dynamics that may eventually lead to irrational behavior. A detailed description for the advantages of using *C. elegans* to study rational decision making is provided in the supplementary information."

In the results, in the context of the newly added behavioral analysis of mutants:

"AWA and AWC had been shown to mediate attraction towards DA^{33,43,45}. Likewise, AWA neurons respond to the stimulant TT⁴⁶, and behavioral assays showed that both AWC and AWA are key for chemotaxis towards TT^{45,47}."

In particular, we added a whole new section of mutant analysis together with a possible mechanism to explain intransitivity (fig. 6) that places *C. elegans* worms at the center of this manuscript (pp. 7-8).

We now discuss *C. elegans* worms in the context of their innate chemotaxis behavior, pages 9-10:

"To construct the comprehensive choice matrix (**Fig. 1**), we used stimuli that are known to attract *C. elegans* worms²⁹. Moreover, some of these stimuli had been shown to be secreted by bacteria, upon which the worms feed^{55,56}. While the concentrations used in our assays may not reflect the typical concentrations found in the natural ecological niches, they still served as potent attractors; thus, the irrationalities reported herein are found in the context of the worms' innate chemotaxis behavior. This crucial feature allowed us to overcome the requirement for initial training procedures that often bias rational behavior studies."

Also in the discussion, we highlight the cross saturation and the behavioral outputs in *C. elegans*. On page 10:

"It was previously shown that the presence of a background stimulus can interfere with chemotaxis towards another stimulus⁴⁵. Here, we found that the presence of an attractive background stimulus not only affected attraction, but actually reversed the preference from attraction to repulsion. *C. elegans* worms may switch preferences from attraction to repulsion when a stimulus concentration increases beyond a certain threshold^{42,45,57}. These repulsive responses often trigger the nociceptive neurons AWB and ASH⁴²; however, our calcium imaging data indicate that activity of these neurons is not modulated between the reciprocal conditions. Instead, we found that AWA, and possibly AWC, are the major players contributing to the DA-mediated TT repulsion. Moreover, the data suggest that TT and DA are sensed by two different receptors on the AWA neurons (**Fig. 6**). Thus, the behavioral switch, and concomitantly the irrational behavior, may largely result from input integration occurring

*within a single type of neurons, AWA, where each input (stimulus) is sensed by its own cognate receptor. Interestingly, such a dual role for a single neuron type was suggested for the AWC neurons, which may mediate both attraction and repulsion from butanone via differential synaptic signaling*⁵⁸. "

These are only numerous examples showing how we refocused the manuscript on *C. elegans* worms.

- The authors summary of their results as "...suggesting that asymmetric representations may provide a simple explanation for irrational behavior" deserves much more explanation and discussion. Are the authors claiming that asymmetric representations explain human irrational behavior? If so, how?

Thank you. We now clarify this point and explicitly explain what we meant in the discussion. We agree that due to the vast differences between mammalian brains and the *C. elegans* neural network, irrationality in higher brain systems is likely to involve various brain regions with more complex neural dynamics. However, we speculate that the same asymmetric modulatory effects between options may underlie irrationality in various brain systems. For example, the famous *asymmetric dominance effect*, which is extensively studied in the context of human decision making, can be viewed as analogous to the asymmetric trans modulation that we observe in *C. elegans*.

We now explicitly discuss this on page 11:

"Interestingly, analogous trans-modulatory effects that lead to irrational behavior were observed in humans as well as in other vertebrate and invertebrate animals^{15,25,27,59}. These are known as the asymmetric dominance effect, or the decoy effect, a form of context-dependent choice in which the probability of choosing one of two options is impacted by the introduction of a third weaker option⁶⁰. According to this effect, adding to a set of options, A and B, a third option C that is asymmetrically dominated by option A (but not by option B), will increase the preference of choosing A over B. In humans, this effect had been extensively documented in various fields, including marketing and consumption⁶¹, policy decisions⁶², and partner preferences⁶³.

*Obviously, mammalian brains are organized completely differently than the *C. elegans* neural network, and irrationality may arise due to complex integrated dynamics within different brain regions, rather than in individual perceptual neurons. Yet, it is possible that the same principles of asymmetric cross- or trans-modulatory effects between options may serve as the underlying basis that leads to irrational behavior."*

Reviewers' Comments:

Reviewer #1:

Remarks to the Author:

The authors have addressed all of my major and minor concerns and the manuscript seems to be much improved for the additional information included based on my recommendations.

George Kemenes

Reviewer #2:

Remarks to the Author:

The revised manuscript by Iwanir et al. is greatly improved from the initial submission. The authors have done a commendably thorough job of addressing the reviewers' concerns. The additional clarifications, statistical analyses, and new data result in a much stronger manuscript. In particular, the new Figure 6 leverages the genetic toolkit of *C. elegans* to address the mechanism by which asymmetric responses can arise; this is an important new addition to the manuscript. The discussion section of the manuscript is also greatly improved. At this point, I have only a few minor comments.

Minor comments:

- 1) The summary is still very jargony, and may deter people from reading the paper.
- 2) I'm not convinced that the supplemental text ("Advantages of *C. elegans*" and "Options as information") adds much to the manuscript.
- 3) In Figure S2, the legend refers to different experiment numbers, but the graphs in the figure aren't numbered, so you don't know which is which.

Reviewer #3:

Remarks to the Author:

The authors have adequately addressed most of my comments. I am still concerned that the overall framing of the manuscript is in the context of human decision-making. For example, the last sentence of the abstract is "Thus, asymmetric modulations between options' representations may provide a simple explanation for irrational behavior". While it's conceivable that asymmetric modulation of representations has some parallel in human decision making, it's hard to imagine this could be a *simple* mechanism for anything, least of all human irrational behavior. (To the authors' credit, the revised discussion section does include a more nuanced summary of their findings.)

A point by point reply to reviewer's comments

Reviewer #1 (Remarks to the Author):

The authors have addressed all of my major and minor concerns and the manuscript seems to be much improved for the additional information included based on my recommendations.

George Kemenes

Thank you for the helpful comments and the kind words.

Reviewer #2 (Remarks to the Author):

The revised manuscript by Iwanir et al. is greatly improved from the initial submission. The authors have done a commendably thorough job of addressing the reviewers' concerns. The additional clarifications, statistical analyses, and new data result in a much stronger manuscript. In particular, the new Figure 6 leverages the genetic toolkit of *C. elegans* to address the mechanism by which asymmetric responses can arise; this is an important new addition to the manuscript. The discussion section of the manuscript is also greatly improved. At this point, I have only a few minor comments.

Indeed, thank you for the kind words and the helpful comments that greatly improved the manuscript.

Minor comments:

1) The summary is still very jargony, and may deter people from reading the paper.

We have now revised the summary and substantially reduced the jargony voice.

2) I'm not convinced that the supplemental text ("Advantages of *C. elegans*" and "Options as information") adds much to the manuscript.

We agree and therefore removed these sections.

3) In Figure S2, the legend refers to different experiment numbers, but the graphs in the figure aren't numbered, so you don't know which is which.

Thank you. We have now numbered the experiments (graph panels) accordingly.

Reviewer #3 (Remarks to the Author):

The authors have adequately addressed most of my comments. I am still concerned that the overall framing of the manuscript is in the context of human decision-making. For example, the last sentence of the abstract is "Thus, asymmetric modulations between options' representations may provide a simple explanation for irrational behavior". While it's conceivable that asymmetric modulation of representations has some parallel in human decision making, it's hard to imagine this could be a *simple* mechanism for anything, least of all human irrational behavior. (To the authors' credit, the revised discussion section does include a more nuanced summary of their findings.)

Thank you.

Following this comment, we have substantially re-framed the manuscript and avoided direct interpretations that relate to human decision making. We have substantially modified the summary to focus on our novel findings as related to the worms' decision making. Accordingly, we modified the introduction by emphasizing *C. elegans* neuro-physiology/ethology studies while toning down rational decision making aspects in higher organisms.